# Decoding long-term trends in the wet deposition of sulfate, nitrate and ammonium after reducing the perturbation from climate anomalies

Xiaohong Yao[1], Leiming Zhang[2]

[1]Key Lab of Marine Environmental Science and Ecology, Ocean University of China, Qingdao 266100, China

[2]Air Quality Research Division, Science and Technology Branch, Environment and Climate Change Canada, Toronto, Canada

Correspondence to: X. Yao (xhyao@ouc.edu.cn) and L. Zhang (leiming.zhang@canada.ca)

**Abstract.** Long-term trends of wet deposition of inorganic ions are affected by multiple factors, among which emission changes and climate conditions are dominant ones. To assess the effectiveness of emission reductions on the wet deposition of pollutants of interest, contributions from these factors to the long-term trends of wet deposition must be isolated. For this purpose, a two-step approach for preprocessing wet deposition data is presented herein. This new approach aims to reduce the impact of climate anomalies on the trend analysis so that the impact of emission reductions on the wet deposition can be revealed. This approach is applied to a two-decade wet deposition dataset of sulfate ($SO_4^{2-}$), nitrate ($NO_3^-$) and ammonium ($NH_4^+$) at rural Canadian sites. Analysis results show that the approach allows for statistically identifying inflection points on decreasing trends in the wet deposition fluxes of $SO_4^{2-}$ and $NO_3^-$ in northern Ontario and Québec. The inflection points match well with the three-phase mitigation of $SO_2$ emissions and two-phase mitigation of NOx emissions in Ontario. Improved correlations between the wet deposition of ions and their precursors' emissions were obtained after reducing the impact from climate anomalies. Furthermore, decadal climate anomalies were identified as dominating the decreasing trends in the wet deposition fluxes of $SO_4^{2-}$ and $NO_3^-$ at a western coastal site. Long-term variations in $NH_4^+$ wet deposition showed no clear trends due to the compensating effects between $NH_3$ emissions, climate anomalies, and chemistry associated with the emission changes of sulfur and nitrogen.

## 1. Introduction

To assess the long-term impacts of acidifying pollutants on the environment, the wet deposition of sulfate ($SO_4^{2-}$), nitrate ($NO_3^-$) and ammonium ($NH_4^+$), among other inorganic ions, has been measured for several decades through monitoring networks

such as the European Monitoring and Evaluation Programme (EMEP) (Fowler et al., 2005, 2007; Rogora et al., 2004, 2016), the National Atmospheric Deposition Program/National Trends Network in the U.S. (Baumgardner et al., 2002; Lehmann et al., 2007; Sickles & Shadwick, 2015), and the Canadian Air and Precipitation Monitoring Network (CAPMoN) (Vet et al., 2014; Zbieranowski and Aherne, 2011). The high-quality data collected from these networks have been widely used to quantify the atmospheric deposition of acidifying pollutants (Lajtha & Jones, 2013; Lynch et al., 2000; Pihl Karlsson et al., 2011; Strock et al., 2014; Vet et al., 2014). The data have also been utilized to identify trends in the atmospheric deposition of reactive nitrogen (Fagerli & Aas, 2008; Fowler et al., 2007; Lehmann et al., 2007; Zbieranowski and Aherne, 2011) and to examine the impacts of acid rain and the perturbation of the natural nitrogen cycle on sensitive ecosystems (Wright et al., 2018). The long-term data can also be used for assessing the effectiveness of environmental policies (Butler et al., 2005; Li et al., 2016; Lloret & Valiela, 2016).

The wet deposition of $SO_4^{2-}$, $NO_3^-$ and $NH_4^+$ is affected by not only their gaseous precursors' emissions (Butler et al., 2005; Fowler et al., 2007; Li et al., 2016) but also complex atmospheric processes such as long-range transport, chemical transformation, and dry and wet removal (Cheng & Zhang, 2017; Yao & Zhang, 2012; Zhang et al., 2012). These processes can be largely affected by climate anomalies. For example, climate anomalies can sometimes bring extreme precipitation amounts in a particular month and subsequently lead to extremely high wet deposition fluxes of ions through enhanced wet removal of air pollutants.. Furthermore, climate anomalies can alter the relative contributions of local sources versus long-range transport to the total wet deposition amounts at reception sites, thereby complicating the relationships between

wet deposition and the emission of air pollutants of interest (Lloret & Valiela, 2016;
Monteith et al., 2016; Pleijel et al., 2016; Wetherbee & Mast, 2016). The emissions of
$SO_2$ and NOx have been decreasing substantially in Europe and North America (Butler
et al., 2005; Li et al., 2016; Pihl Karlsson et al., 2011); coincidently, climate anomalies
have also occurred more frequently in the recent decades (Burakowski et al., 2008;
Lloret & Valiela, 2016; Wijngaard et al., 2003), thereby leading to more complicated
linkages between wet deposition and emission trends on decadal scales.

Many trend analysis studies in the literature simply examined annual or seasonal values
as the data inputs for two popular trend analysis tools, i.e., the Mann-Kendall (M-K)
and linear regression (LR) methods (Marchetto et al., 2013; Waldner et al., 2014 and
references therein). These studies focused on the detection of statistically significant
trends; for example, Waldner et al. (2014) conducted a comprehensive analysis on the
applicability of the techniques to different choices of length and temporal resolutions
of a data series. Regarding the resolved trend results, these approaches are not well
suited to separating the impact of air pollutants' mitigation from the perturbation by
climate anomalies. Large uncertainties thus existed in the studies interpreting the major
driving forces determining the extracted trends in the wet deposition of $SO_4^{2-}$, $NO_3^-$ and
$NH_4^+$. Regarding that air pollutant's emission mitigation targets often vary in different
phases of the entire study period, inflection points may exist in the trends in the wet
deposition of ions. The inflection points were rarely studied, despite their importance
for assessing the effectiveness of environmental policies. An alternative would be to
use high time resolution data in the Ensemble Empirical Mode Decomposition (EEMD)
method (Wu & Huang, 2009); however, this method still suffers from the end effect in
certain scenarios, whereby the extracted trends cannot be explained (Yao & Zhang,

76    2016).


A new approach is presented herein that aims to reduce the perturbations from climate
anomalies on data inputs so that robust trends can be elucidated for evaluating the
effectiveness of emission control policies. In this approach, raw data are preprocessed
to generate a new variable, which is then applied to M-K and LR methods. A piecewise
linear regression (PLR) is also used to extract trends for cases in presence of inflection
points. The extracted trends in the wet deposition data on a decadal scale are then
properly linked to major driving forces such as emission reductions and climate
anomalies. This new approach is first applied to the wet deposition data of $SO_4^{2-}$, $NO_3^-$
and $NH_4^+$ in Canada, as an example to demonstrate its capability and advantages over
the traditional approaches. The extracted trends in the wet deposition of ions are further
studied through correlation analysis with known emission trends of their respective
gaseous precursors ($SO_2$, NOx and $NH_3$) in Canada and the U.S. Major driving forces
for the trends of ion wet deposition and how the wet deposition ions responded to their
precursors' emissions in Canada are then revealed.

**2.    Methodology**
*2.1 Data sources*
Wet        deposition        flux        ($F_{wet}$)        data        were        obtained        from        CAPMoN
(https://www.canada.ca/en/environment-climate-change/services/air-
pollution/monitoring-networks-data/canadian-air-precipitation.html). Data from four
sites have been collected for over twenty years and were chosen herein to illustrate the
novel trend analysis method (Table S1). Site 1 is an inland forest site at Chapais in
Québec. Site 2 is situated in a coastal forest area at Saturna in British Columbia. Sites
3 and 4 are two inland forest sites at the Chalk River and at Algoma, respectively, in
northern Ontario. Details on data sampling, chemical analysis and quality control can
be found in previous studies (Cheng & Zhang, 2017; Vet & Ro, 2008; Vet et al., 2014).
The emissions data of gaseous precursors were downloaded from the Air Pollutant
Emission Inventory (APEI, https://pollution-waste.canada.ca/air-emission-inventory/)
in   Canada   and   from   the   USEPA   National   Emissions   Inventory   (NEI,
https://www.epa.gov/air-emissions-inventories/air-emissions-sources)   in   the   U.S.
These data were demarcated at a provincial level in Canada and at a state level in the
U.S. Data for the years of 1990 to 2011, which correspond to the period of selected $F_{wet}$
data, were used in this study.

*2.2 Statistical methods*
The M-K method is a popular nonparametric statistical procedure that can yield
qualitative trend results, such as "an increasing/decreasing trend with a P value of
<0.05," "a probable increasing/decreasing trend with a P value of 0.05-0.1," "a stable
trend with a P value of >0.1, as well as a ratio of <1.0 between the standard deviation
and the mean of the dataset," and "a no trend for P>0.1 with all other conditions"
(Kampata et al., 2008; Marchetto et al., 2013). The LR method has also been widely
used to extract trends (Marchetto et al., 2013; Waldner et al., 2014). Zbieranowski and
Aherne (2011) used LR to extract trends by separating different phases because of the
presence of inflection points in the entire study period, and the approach is same as PLR
(Vieth, 1989).  In this study, the three methods were employed to compute the trends
of   ion   wet   deposition   using   software   downloaded   from   https://www.gsi-
net.com/en/software/free-software/gsi-mann-kendall-toolkit.html and Excel 2016, first
using the annual $F_{wet}$ directly as input data, then using a modified input data set, as
described in Section 2.3.

The annual $F_{wet}$ is widely used for trend analysis and the trend results are thereby used
to compare with those derived from the approach proposed in this study. Note that $R^2$
is conventionally used in LR and PRL. However, r instead of $R^2$ is used in correlation
analysis. Thus, $R^2$ and r are used for the two types of analyses in this study, respectively
Moreover, several methods can be used to do PRL in classical statistics literature. The
simplest one is to manually conduct piecewise regression where inflection points are
visible to be recognized, and this approach is used in this study. More complex
algorithms are also available in literature to conduct PRL for datasets with hundreds of
points (Ryan and Porth, 2007 and references therein). The complex algorithms have
seldom been used to identify trends in annul wet deposition of ions because of the short
data record.

*2.3 Filtering climate anomalies*
The modified input data set was produced in two steps. The first step was an effort to
reduce the perturbation from the monthly climate anomalies to the input data. This was
done by creating a new variable that was defined as the slopes of the regression
equations of a series of study years against a climatology (base) year using monthly
$F_{wet}$ data. Note that the monthly $F_{wet}$ data were aggregated from daily raw data before
the regression analysis. To ensure the presence of enough data points in each regression
equation, the data corresponding to two-year periods (or 24 monthly $F_{wet}$ values) were
grouped together, as detailed below. At a selected site and for a given chemical
component, monthly $F_{wet}$ data were generated for the first two years and were grouped
together and rearranged from the smallest to the largest values to form an array of data
with 24 data points, i.e., A(i) with i=1 to 24. Repeating the above procedure for the
subsequent years using a two-year interval to eventually obtain a series of data arrays,
A(i) now becomes A(i, j) with i=1 to 24 and j=1 to N, where N is the total number of
data arrays. The climatology data array (CA(i)) was then defined as the average of all
of the arrays as follows:
$CA(i) = \frac{1}{N}\sum_{j=1}^{N} \quad A(i,j), \; i = 1 \; to \; 24.$

LR with zero interception was applied for each individual data array against the
climatology data array. In cases where the maximum monthly deposition flux deviated
greatly from the general regression curve, the slopes (m-values) were calculated after
excluding the maximum monthly deposition flux, which is an approach that reduced
the perturbation to the m-values from the monthly scale climate anomalies. The second
step was to screen out the outliers in m-values, which reduced the perturbation to the
m-values from the annual-scale climate anomalies.

*2.4 Example case for data filtering*
An analysis of Site 1 is used to illustrate the new approach and demonstrate its
advantages against the existing common approaches used in the literature. Twelve two-
year periods of data (1988-1989, 1990-1991, etc.) are available from this site. The
regression of each data set against the climatology data set was first performed using
all of the monthly values to obtain an m-value (the slope) (Fig. 1a-d). For eight out of
the 12 data sets, the m-values were recalculated after excluding the maximum monthly
value of $F_{wet}$, which appeared to be an apparent outlier of the linear regression. Three
out of the 12 data sets showed the maximum $F_{wet}$ being positively deviated from the
general trend, five negatively deviated from the general trend, and four consistent with
the general trend. The $R^2$ values were then significantly increased for the eight sets,
e.g., from the original values of 0.79-0.94 to the improved values of 0.92-0.98. To
demonstrate that the excluded maximum value was an outlier, the case of the 1990-
1991 data set was taken as an example. The new regression equation (y=1.47x, $R^2$=0.98,
Fig. 1a) predicted a maximum value in the range of 330-368 mg m$^{-2}$ month$^{-1}$ using three
times the standard deviation ($\pm 3$ SD, 0.08) at a 99% confidence level. The actual
observed maximum value of 532 mg m$^{-2}$ month$^{-1}$ was much larger than the upper range
of the predicted value and was thus believed to be caused by monthly scale climate
anomalies, i.e., the occurrence of extreme amount of precipitation. The maximum
monthly deposition flux in 1990-1991 occurred in September 1990 when the monthly
precipitation depth reached 294 mm, which was much higher than those in the same
month of other years, e.g., 169, 68, 95 and 127 mm in 1988, 1989, 1991 and 1992,
respectively. The maximum daily precipitation depth in September was also higher in
1990 (91 mm) than in other years (43.6, 12.2, 13.6 and 26.8 mm in 1988, 1989, 1991
and 1992, respectively). However, the monthly geometric average concentration of
$SO_4^{2-}$ in precipitation (1.8 mg L$^{-1}$) in September 1990 was close to the mean value
($1.7\pm 0.3$ mg L$^{-1}$) in September 1988-1992 and  was even smaller than that (2.9 mg L$^{-1}$)
in August 1990. The maximum value was treated as an outlier and excluded for
analysis.

Using the similar procedure, all outliers in this study were identified. The exclusion of
the observed maximum value greatly reduced the perturbation of the short-term climate
anomalies to the calculated m-value in this two-year period, i.e., the m-value decreased
from 1.67 to 1.47, which in turn increased the relative contribution of the air pollutants'
emissions to the calculated m-value. Note that monthly changes in emissions may not
impact the $F_{wet}$ as much as does a large monthly change in precipitation depth or
concentration in precipitation. For example, the monthly average concentrations of $SO_2$
were almost the same in May, September and October of 1990 (~0.7 µg m$^{-3}$) while the
monthly $F_{wet}$ of $SO_4^{2-}$ varied significantly, e.g., 113, 179 and 532 mg m$^{-2}$ month$^{-1}$ ,
respectively in the same months. The monthly average concentration of $SO_2$ in February
(4.8 µg m$^{-3}$) was the largest among the twelve months of 1990, but the corresponding
monthly $F_{wet}$ of $SO_4^{2-}$ was the smallest (34 mg m$^{-2}$ month$^{-1}$).

Even through comprehensive analysis, any single climate factor alone, including
monthly precipitation depth, was apparently unable to explain the negative deviation of
the maximum monthly value of $F_{wet}$ from the general trend. The causes of such a
negative deviation is yet to be identified. In summary, the new approach proposed
above by applying the criteria of being outside the boundaries of ±3 times the standard
deviation of the general trend meets the objective of identifying outlier data points.

The revised m-values were further scrutinized by eliminating the outliers caused by the
annual-scale climate anomalies. For example, the m-value of 1.31 in 1998-1999 greatly
deviated from other m-values, narrowly oscillating approximately 0.96±0.07 (average
± 1 SD) during the period of 1994-2005, even with the ±3 SD being considered (Fig.
1a-d). Using the value of 0.96 as the reference, climate anomalies likely increased the
$F_{wet}$ of $SO_4^{2-}$ by 37% in 1998-1999. The m-values were then calculated by shifting one
year in time to 1997-1998 (1.07) and to 1999-2000 (1.24). The $F_{wet}$ in 1998 was less
affected by climate anomalies than that in 1999. Thus, the m-value in 1997-1998 was
within 0.96±0.21 (average ± 3 SD) and used to replace the m-value in 1998-1999 for
the trend analysis. Similar to the first step discussed above, this approach meets the
objective of identifying outlier m-values by applying the criteria of being outside the
range of ±3 SD plus the average m-value during a decade or a longer period. The
abnormally increased $F_{wet}$ of $SO_4^{2-}$ in 1999 was mainly because of the increased
precipitation depth (1312 mm), which was the largest during 1998-2011 (the annual
average precipitation depth excluding 1999 was 1067±86 mm). However, the geometric
average concentration of $SO_4^{2-}$ in precipitation in 1999 (1.0 mg L$^{-1}$) was close to those
in the other years, e.g., 0.9 mg L$^{-1}$ in 1997 and 1998 and 1.0 mg L$^{-1}$ in 2000.

*2.5 Justification for the new approach* More justification of the new approach can be
found in the Supporting Information, including Figs. S1-6, wherein the statistical
comparison between this and other approaches was presented. Theoretically, the
extracted trend using the data preprocessed with the new approach is determined by the
local emissions of air pollutants, the regional transport of air pollutants, and climate
anomalies that are unable to be removed by the new approach. It is assumed that the
extracted trend is less affected by microphysical/chemical processes, since two-year
data were used together to calculate the m-value.

In theory, if the data from different sites in the same region are grouped together for
trend analysis, the results may be better linked to the trends of the regional emissions
of related air pollutants. In the following sections, trend analysis results from individual
sites as well as those from grouped sites are discussed. Sites 1, 3 and 4 showed similar
trends in the wet deposition of $SO_4^{2-}$ and $NO_3^-$, and these three sites were grouped
together.

**3. Results and discussion**
*3.1 Trends at Site 1 after reducing perturbations from climate anomalies*
Trends in the m-values shown in Fig. 2 represent the trends after removing the
perturbations from climate anomalies at Site 1 in northern Québec from 1988 to 2011.
$SO_4^{2-}$ and $NO_3^{-}$ showed decreasing trends from a LR analysis, with $R^2$ values of 0.81
and 0.71, respectively, and P values <0.01 (Fig. 2a and 2d). The decreasing trends were
also confirmed by the M-K method analysis. $NH_4^{+}$ exhibited a stable trend from M-K
analysis (Fig. 2g), as well as no significant trend with P value >0.05 from LR analysis.
The annual $F_{wet}$ of these ions are also shown in Figs. 2b, 2e and 2f and annual emissions
of $SO_2$, $NO_x$ and $NH_3$ in Figs. 2c, 2f and 2i, respectively. These data were used to
compare and facilitate analysis in terms of identifying inflection points and the
advantage of using the m-value over the annual $F_{wet}$, as presented below.

The m-values of $SO_4^{2-}$ and $NO_3^{-}$ also allowed for statistical identification of trends in
different phases supported by annual variations in emissions of $SO_2$ and $NO_x$ (Figs. 2c
and 2f) to some extent. The inflection point for each phase is critical to a) link the annual
$F_{wet}$ of ions and the emissions of the corresponding precursors, and b) assess the
effectiveness of environmental policies. For example, the trends in the m-values of
$SO_4^{2-}$ can be clearly classified into three phases (Fig. 2a). Therefore, PLR should be
applied separately for the different phases in the presence of the inflection points, rather
than LR for the entire period, and the result is presented as:
$$\begin{cases} m - value = 1.38, 1988 \leq x < 1994 \\ m - value = 1.02, 1994 \leq x \leq 2005 \\ m - value = -0.185 * \left(\frac{x}{2} - 1001\right) + 1.15, 2005 < x \leq 2010 \end{cases}$$

where x represents the calendar year from 1988 to 2010.
The m-values oscillated approximately 1.38±0.08 during Phase 1 (1988 to 1993) and
approximately 1.02±0.08 during Phase 2 (1994 to 2005), with a significant difference
between the two phases under the t-test (P value <0.01), thereby implying an abrupt
decrease of approximately 30% at the inflection point between the two phases. The m-
values linearly decreased by approximately 20% every two years, starting from the end
of Phase 2 to Phase 3 (2006-2011). Again, a significant difference existed between
Phase 2 and Phase 3 under the t-test (P value <0.01). The three phases generally aligned
with the three-phase regulated $SO_2$ emissions in Ontario. It should be stated that Phase
1 and Phase 3 each covered only six years (N=6). Cautions should be taken to explain
the trend result in each phase in relation to precursors' emissions.

The PRL result of $NO_3^-$ is expressed as:
$$\begin{cases} m - value = 1.09, 1988 \le x < 2004 \\ m - value = -0.128 * \left(\frac{x}{2} - 1001\right) + 1.08, 2004 \le x \le 2010 \end{cases}$$

The trend in the m-values of $NO_3^-$ can be classified into two phases with the inflection
point at 2003, which was confirmed by the t-test result, i.e., the values oscillated
approximately 1.09±0.09 during the period from 1988 to 2003 and then exhibited a
significant decrease of approximately 50% overall afterwards, with P value <0.01.
The m-value of $NO_3^-$ in 1998-1999 was approximately 30% larger than the mean value
in 1988-2003 and exceeded the mean value plus 3 SD in 1998-2003, and thus was not
included in the trend analysis. The sharp increase in $F_{wet}$ of $NO_3^-$ occurred mainly in
1999, which was probably due to largely increased annual precipitation depth as
mentioned in Section 2.4. The analysis was also supported by the geometric average
concentration of $NO_3^-$ in precipitation, which was 1.1 mg $L^{-1}$ in 1999, 5% lower than
that in 1988 and only 5-10% higher than those in 1990-1991, 1993 and 2002.
Moreover, the monthly $F_{wet}$ values of $NO_3^-$ in March, April, July and August 1999 were
actually lower than the corresponding long-term averages in 1988-2003 (excluding
1999) (Fig. S6a). This outcome indicates that the large increase in annual $F_{wet}$ of $NO_3^-$
in 1999 was unlikely to have been determined by the emissions of its gaseous
precursors. The same can be said for the large increase in $F_{wet}$ of $SO_4^{2-}$ in 1999 (Fig. 2a,
S6b).

To demonstrate the advantage of using the m-values in trend analysis, m-values were
correlated to the reported emissions of concerned air pollutants. The trends in the m-
value of $SO_4^{2-}$ at Site 1 (Fig. 2a) were clearly different from those of the $SO_2$ emissions
in Québec (Fig. 2c) but matched well to those in Ontario (Fig. 2c), which is also
supported by their Pearson correlation coefficients, e.g., no significant correlation (r =
0.46 and P value >0.05) for the former case and a good correlation (r = 0.96 and P value
<0.01) for the latter case. Zhang et al. (2008) reported that this remote area can receive
the long-range transport of air pollutants from Ontario but that transport is less likely
from the intensive emission sources in Québec.

In addition, LR analysis of the annual $F_{wet}$ of $SO_4^{2-}$ revealed a decreasing trend (second
row in Fig. 2b). The M-K method analysis also confirmed the decreasing trend with
annual $F_{wet}$ as input. However, the three-phase trend in $F_{wet}$ of $SO_4^{2-}$ and related
inflection points, identified using the m-values discussed above, were not identified by
the t-test when simply using annual $F_{wet}$ data as input. Identifying these inflection points
is crucial to assess the effectiveness of environmental policies. The correlation between
annual $F_{wet}$ and emission was 0.89 for $SO_4^{2-}$ vs. $SO_2$ in Ontario (P values <0.01), while
the corresponding r value was as high as 0.96 between m-value and emission. After
reducing the perturbations from climatic factors to the annual $F_{wet}$, a stronger
correlation was obtained between $F_{wet}$ and emission. The increased r further solidified
the dominant contribution of the long-range transport of air pollutants from Ontario
rather than Québec to the wet deposition of $SO_4^{2-}$ at Site 1.

The trends in NOx emissions during 1990-2003 had similar bell-shape patterns in
Québec and Ontario, although with different magnitudes of emissions (Fig. 2f). A
different trend pattern was seen for the m-value of $NO_3^-$ at Site 1 than for the
abovementioned provincial emissions during the same period (Fig. 2d), and there was
no significant correlation (r<0.41, with P value >0.05) between the m-value of $NO_3^-$
and the emissions of NOx in Québec or Ontario. Different results were found for the
period of 2002-2011 than those of 1990-2003 discussed above. In 2002-2011, the m-
value of $NO_3^-$ decreased by ~50% and the NOx emissions decreased by ~40% in
Québec and Ontario; also, good correlations (r = 0.94-0.95 with P values <0.01) were
observed between m-values and emissions. The contrasting correlation results between
the two different periods discussed above implied the complex link between wet
deposition of $NO_3^-$ and emissions of $NO_x$. One might assume that the perturbation from
climate anomalies might not be fully removed by the new approach for the period of
1990-2003, which overwhelmed the effects of NOx emissions on the trends in m-values
of $NO_3^-$. Such a possibility is practically very low since the approach works well for the
period of 2002-2011. The contrasting results between these two periods are yet to be
explained. $F_{wet}$ of $NO_3^-$ and precipitation depth exhibited only a weakly significant
correlation, with r = 0.58 and P<0.05 in 1988-2003 (the values in 1999 were excluded).
Annual precipitation varied by only ~20% during the fifteen years, and this factor alone
was unlikely to explain the ~100% interannual variation of $F_{wet}$ of $NO_3^-$ during that
period.

LR analysis of the annual $F_{wet}$ of $NO_3^-$ revealed a decreasing trend (second row in Fig.
2e), confirmed by the M-K method analysis. However, the two-phase trend in $F_{wet}$ of
$NO_3^-$ and related inflection point were not identified by the t-test when simply using
annual $F_{wet}$ data as input. The correlations between annual $F_{wet}$ and emission were 0.74-
0.76 for $NO_3^-$ vs. $NO_x$ in Québec and Ontario (P values <0.01), while the corresponding
r values increased to 0.84-0.85 between m-value and emission. Both the identified
inflection point and the stronger correlation between m-value and emission
demonstrated the advantage of using the m-value over annual $F_{wet}$ of $NO_3^-$ in trend
analysis.

The m-value of $NH_4^+$ at Site 1 had no significant correlation (r = 0.21 and P value >0.05)
with the emission of $NH_3$ in Québec but exhibited a weakly significant correlation (r =
0.60 and P value <0.05) with the emission of $NH_3$ in Ontario. Nearly all of the $NH_4^+$
was associated with $SO_4^{2-}$ and $NO_3^-$ in the atmosphere (Cheng and Zhang, 2017; Teng
et al., 2017; Tost et al., 2007; Zhang et al., 2012), and the trends in the m-value of $NH_4^+$
could be affected by many other factors besides $NH_3$ emissions and climate anomalies,
e.g., gas-aerosol partitioning and different dry and wet removal efficiencies between
$NH_3$ and $NH_4^+$, pH value of wet deposition.


The stable trend in annual $F_{wet}$ of $NH_4^+$ and the decreasing trend in annual $F_{wet}$ of $NO_3^-$
gradually increased the relative contributions of reduced nitrogen in the total nitrogen
wet deposition budget, e.g., from 40% in 1998-1999 to 52% in 2010-2011. A similar
trend has also been recently reported in the U.S. (Li et al., 2016). Such a trend was
mostly due to the mitigation of NOx rather than climate anomalies.

*3.2 Decadal climate anomalies drove trends at Site 2*

3.2.1 Trends in m-value of $SO_4^{2-}$

Fig. 3 shows the trend analysis results at Site 2. An obvious shift in the m-values and annual $F_{wet}$ occurred during 2001-2002, as detected by the t-test, i.e., the m-values of $SO_4^{2-}$ oscillated approximately 1.15±0.11 in 1990-2001 and 0.76±0.02 in 2002-2011 (or 0.83±0.12 if the value in 2006-2007 was included), but with a significant difference between the two periods with P value <0.01. The annual $F_{wet}$ of $SO_4^{2-}$ oscillated approximately 632±63 mg m$^{-2}$ in 1990-2001 and 452±74 mg m$^{-2}$ in 2002-2011, and the values between the two periods showed significant differences. The shift led to the m-values and annual $F_{wet}$ of $SO_4^{2-}$ exhibiting a consistent decreasing trend by ~40% overall from 1990 to 2011 using the LR and the M-K method.

The emissions of $SO_2$ oscillated approximately 1.13±0.07 in 1990-2001 and 1.06±0.03 in 2002-2011 in British Columbia, which did not support the large decrease of approximately 40% in wet deposition of $SO_4^{2-}$ in 2002-2011. Statistically, no correlation existed between annual $F_{wet}$ of $SO_4^{2-}$ and the emissions of $SO_2$ in British Columbia, with r = 0.52 and P value >0.05. Although the transboundary transport of air pollutants from the U.S. cannot be excluded, the almost constant m-values from 2002 to 2011 (excluding 2006-2007) at Site 2 were inconsistent with the approximately 70% decrease in emissions of $SO_2$ in the state of Washington in the U.S. during that period (not shown). Precipitation cannot explain the jump in wet deposition either, because there was no corresponding jump in precipitation during 2001-2002 (Fig. 3b).

van Donkelaar et al. (2008) analyzed aircraft and satellite measurements from the

Intercontinental Chemical Transport Experiment and proposed the long-range transport
of sulfur from East Asia to the west coast of Canada. The wind vector and wind speed
from the North American Regional Reanalysis (NARR), with a spatial resolution of 32
km by 32 km (Mesinger et al., 2006), were thereby analyzed to study the decadal
changes in wind fields and associated potential impacts on the long-range transport of
air pollutants over the western coastal Canada and U.S. The average wind fields
including mean wind vector and speed (shading in Fig 4a-d) in 1990-2011 at 925 hPa
showed air masses over the western coastal Canada and U.S. were primarily originated
from the Pacific Ocean (Fig. 4a). However, the anomalies of wind fields in 1990-2001
relative to 1990-2009 clearly showed a counterclockwise pattern in the corresponding
coastal area, including Site 2., while a clockwise pattern existed in 2002-2011 relative
to 1990-2009 (Fig. 4b, c). The anomalies shown in Fig. 4c indicated the northwesterly
wind being enhanced in 2002-2011 over the western coastal Canada and U.S., possibly
reducing air pollutants being transported from the continent to Site 2.  In contrast, the
anomalies in Fig. 4b indicated that the northwesterly wind was reduced in 1990-2001.
Consequently, more air pollutants might have been transported from the continent to
Site 2, resulting in a distinct demarcation in 2002. This hypothesis was also supported
by a large rebound of the m-value in 2006-2007, due to the increase in $F_{wet}$ of $SO_4^{2-}$ in
2007. The climate anomalies of wind fields in 2007 relative to 1990-2009 showed a
counterclockwise pattern in the north, while the clockwise pattern was pushed to the
south (Fig. 4d). With the northwesterly wind being reduced, a greater contribution of
air pollutants from the coast of Canada and U.S. to Site 2 might have led to the large
increase in $F_{wet}$ of $SO_4^{2-}$ during a few month-long periods in 2007.

The present study is the first one identifying the decreasing trend in the annual $F_{wet}$ of
$SO_4^{2-}$ as being very likely caused by decadal climate anomalies, i.e., wind fields, rather
than by the emission reductions of $SO_2$. The decadal anomalies of wind fields may
substantially alter the long-range transport of air pollutants to the reception site. Note
that the causes for the decadal anomalies of wind fields in this region are beyond the
scope of the present study, but some information can be found in the literature (Bond
et al., 2003; Coopersmith et al., 2014; Deng et al., 2014).

3.2.2 Trends in m-values of $NO_3^-$ and $NH_4^+$
For the wet deposition of $NO_3^-$, the m-values also showed a clear shift, i.e., the m-values
oscillated approximately 1.09±0.14 in 1990-2001 and 0.88±0.06 in 2002-2011, with a
significant difference between the two periods under the t-test with P value <0.01. The
annual $F_{wet}$ of $NO_3^-$ varied substantially, and the shift could not be identified
statistically. However, the annual $F_{wet}$ of $NO_3^-$ exhibited a decreasing trend by M-K
method analysis. Similar to the case of $SO_4^{2-}$, no significant correlation (r = 0.49, P
value >0.05) existed between the annual $F_{wet}$ of $NO_3^-$ and the emissions of NOx in
British Columbia.

In addition to decadal anomalies of wind fields, the interannual climate variability such
as precipitation depth, annual anomalies of wind fields in 2007, etc., (Fig. 3b) also
affected the trends in m-values and annual $F_{wet}$ of $NO_3^-$. The annual precipitation depth
largely varied from 601 mm to 1054 mm in the two decades. The perturbations from
interannual variability of precipitation depth cannot be completely removed by the new
approach. For example, the calculated m-values in 1992-1993 and 1994-1995 were
evidently lower than the m-values in 1990-2001. However, the annual geometric
average concentrations of $NO_3^-$ in 1992-1995 varied around 0.77±0.11 mg $L^{-1}$ and were
even larger than the values of $0.66\pm0.08$ mg L$^{-1}$ in 1990-2001 (excluding 1992-1995).
The lower m-values were mainly attributed to the lower precipitation depth in 1992-
1994 (Fig 3b) rather than lower emissions of NOx. Interannual climate variability
including precipitation depth and annul anomalies of wind fields may complicate the
relationship between the $F_{wet}$ of $NO_3^-$ and the emissions of $NO_x$ in British Columbia.
For example, the m-values in 1990-1991, 1996-1997, 1998-1999 and 2000-2001 were
nearly constant at $1.17\pm0.03$; however, the NOx emissions in British Columbia in 1998-
1999 were 26% greater than those in 1990-1991. Moreover, there was a sharp decrease
in the NOx emissions (by ~30%) from 2002 to 2011 in British Columbia. However, the
m-values oscillated approximately $0.88\pm0.06$ and showed no clear trend based on either
the M-K method or LR analysis. The interannual climate variability apparently negated
the impact of reduced emissions during these periods.

The m-values and the annual $F_{wet}$ of $NH_4^+$ oscillated approximately $0.99\pm0.13$ and
$81\pm16$ mg m$^{-3}$, respectively, in the period of 1990-2011, and showed no trend (Fig. 3).
Neither the m-values nor annual $F_{wet}$ of $NH_4^+$ showed the two-period distribution
pattern or had any significant correlation with the emissions of $NH_3$ in British Columbia
at a 95% confidence level. Similarly to Site 1, the annual variation in $F_{wet}$ of $NH_4^+$ at
Site 2 cannot be simply explained by known emission trends.

In summary, decadal anomalies of wind fields overwhelmingly determined the long-
term trends in the wet deposition of $SO_4^{2-}$ and $NO_3^-$, with the perturbation from monthly
and annual climate anomalies removed at Site 2. The interannual climate variability
including precipitation depth, annual anomalies of wind fields, etc., further complicated
the trends, resulting in undetectable influences of the emission trends on the deposition
trends. Since the decrease in $F_{wet}$ of $NO_3^-$ appeared to be primarily caused by decadal
climate anomalies of wind fields, the relative contributions of $NH_4^+$ and $NO_3^-$ in the
total N wet deposition varied little, i.e., 33% versus 67% in 2010-2011 and 31% versus
69% in 1990-1991.

*3.3 Regional trends in wet deposition in northern Ontario* and Québec
Trends in the m-values or annual $F_{wet}$ of ions at Sites 3 and 4 in the northern regions of
Ontario were generally similar to those found at Site 1 (Fig. S5 and S6). The three-
phase trend in m-values of $SO_4^{2-}$ and the two-phase trend in m-values of $NO_3^-$ were also
obtained at Sites 3 and 4 after excluding a few m-values that were caused by large
perturbations from climate anomalies. For example, the annul precipitation depths of
1044 mm in 1987 and 905 mm in 1997 at Site 4 were evidently lower than the average
value of 1299±124 mm (excluding 1987 and 1997) in 1985-1997 (Table S2). However,
the geometric average concentration of $SO_4^{2-}$ of 1.5 mg $L^{-1}$ in 1997 was the same as the
mean value of 1.5±0.2 mg $L^{-1}$ in 1995-1999 (excluding 1997). The value of 1.6 mg $L^{-1}$
in 1987 was also same as that in 1989. The lower annul precipitation depths in 1987
and 1997 than in the other years were very likely the dominant factor causing the
abnormally lower m-values in 1986-1987 and 1996-1997. Thus, Sites 1, 3 and 4 were
combined together to study regional trends in the northern areas of Ontario and Québec
(Fig. 5a-c). Similar to those found at the individual sites, the temporal profile of regional
m-values of $SO_4^{2-}$ can be clearly classified into three phases (Fig. 5a) as follows: Phase
1 from 1988 to 1993 with m-values oscillating approximately 1.31±0.08, Phase 2 from
1994 to 2003 with near-constant m-values of 1.05±0.04, and Phase 3 for 2004 onward
with a decreasing trend by an overall ~50%. Significant differences of m-values existed
between any two of the three phases, based on the t-test results (P value <0.01). The
PRL result is expressed as below:

$$\begin{cases} m-value = 1.31, 1988 \leq x < 1994 \\ m-value = 1.05, 1994 \leq x < 2004 \\ m-value = -0.129 * \left(\frac{x}{2} - 1001\right) + 1.03, 2004 \leq x \leq 2010 \end{cases}$$

The three-phase pattern of m-values matched well with the three-phase emission profile
of $SO_2$ in Ontario. Statistically, a ~70% decrease in m-value and a ~70% decrease in
emissions were found from 1990 to 2011, with a correlation of $r = 0.95$ (P value <0.01).

The profile of the regional m-values of $NO_3^-$ also clearly exhibited two phases,
according to the following t-test results: Phase 1 from 1988 to 2003, with m-values
narrowly varying approximately $1.11 \pm 0.05$, and Phase 2 from 2004 to 2011 with a
decreasing trend by an overall ~40% against that in 2002-2003 (Fig. 5b). The PRL
result is expressed as below:

$$\begin{cases} m-value = 1.11, 1988 \leq x < 2004 \\ m-value = -0.11 * \left(\frac{x}{2} - 1001\right) + 1.03, 2004 \leq x \leq 2010 \end{cases}$$

From 2002 to 2011, the m-value had a moderately good correlation with the NOx
emission in Ontario ($r = 0.91$, P<0.01), and the two variables decreased by 30-40% in
this period. From 1990 to 2003, the near constant m-value was, however, inconsistent
with the bell-shape profile of the NOx emissions mainly caused by annual variations in
NOx emission from the sector of Transportation and Mobile Equipment in Ontario and
Québec, which could be due to either the perturbation from climate anomalies or
unrealistic emissions inventory from (APEI) in Canada. Considering that the first
possibility was minimal over a large regional scale, especially when the consistency
was determined in a different time frame (2002-2011) in the same region, it is thus
doubtful that the bell-shape profile of the NOx emissions in 1990-2003 was realistic.

The regional m-values of $NH_4^+$ largely oscillated from 1988 to 2003 (Fig. 5c). The m-
values of $NH_4^+$, however, decreased by ~30% from 2002 to 2011, leading to a probable
decreasing trend in m-value from 1988 to 2011. No correlation was found between the
m-values of $NH_4^+$ and the emissions of $NH_3$ in Ontario, which is consistent with the
findings at the individual sites discussed above.

Since the decrease in $F_{wet}$ values of $NO_3^-$ at Sites 3 and 4 were very likely due to the
mitigation of NOx in Ontario, the decrease also changed the relative contributions
between $NH_4^+$ and $NO_3^-$ in the total N wet deposition budget. For example, $NH_4^+$ and
$NO_3^-$ contributed 52% and 48%, respectively, to the total budget in 2010-2011 and 34%
and 66%, respectively, in 1984-1985 at Site 3. The corresponding numbers at Site 4
were 58% and 42% in 2010-2011 and 47% and 53% in 1985-1986.

**4. *Conclusions***
Climate anomalies during the two-decade period resulted in annual $F_{wet}$ of $SO_4^{2-}$ and/or
$NO_3^-$ deviating from the normal value by up to ~40% at the rural Canadian sites. The
new approach of rearranging and screening $F_{wet}$ data can largely reduce the impact of
climate anomalies when used for generating the decadal trends of $F_{wet}$. With the climate
perturbation being reduced, $F_{wet}$ of $SO_4^{2-}$ exhibited a three-phase decreasing trend at
every individual site, as well as on a regional scale in northern Ontario and Québec.
The three-phase pattern of the decreasing trend in $F_{wet}$ of $SO_4^{2-}$ matches well with the
emission trends of $SO_2$ in Ontario, as supported by the good correlation between wet
deposition and emission, with r $\geq$0.95 and P<0.01. $F_{wet}$ of $NO_3^-$ exhibited a two-phase
decreasing trend, but only during the second phase $F_{wet}$ of $NO_3^-$, and the emissions of
NOx in Ontario and Québec matched well, with a good correlation of r $\geq$0.91 and
P<0.01. Compared to the results obtained without applying the new approach, it is
concluded that, after reducing the perturbation from climate anomalies, 1) better
correlation was obtained between $F_{wet}$ of ions and the emission of the corresponding
gaseous precursors in northern Ontario and Québec, and 2) the inflection points in the
decreasing trends of $F_{wet}$ of $SO_4^{2-}$ and $NO_3^-$ were visibly and statistically identified.

However, the new approach cannot completely remove the perturbations from climate
anomalies, especially when this is the dominant factor and/or on long timescales, as
was the case at a coastal site of Saturna in British Columbia. At this location, the
decreasing trends in $F_{wet}$ of $SO_4^{2-}$ and $NO_3^-$ were caused by the decadal anomalies of
wind fields, as well as being affected by interannual climate variability including
precipitation depth and annul anomalies of wind fields, etc., which overwhelmed the
impact of the emission changes of the gaseous precursors in this province. This is the
first study that has identified that decadal anomalies of wind fields can dominate trends
in $F_{wet}$ of $SO_4^{2-}$ and $NO_3^-$. The new findings will stimulate more studies on the impacts
of decadal climate anomalies on atmospheric deposition of concerned air pollutants.
The long-term variations in $F_{wet}$ of $NH_4^+$ generally showed no clear long-term trends.
Moreover, no apparent cause-effect relationships were found between the wet
deposition of $NH_4^+$ and the emission of $NH_3$. It can be reasonably inferred that
additional key factors besides those discussed in this study also impact the trends of
$F_{wet}$ of $NH_4^+$. Thus, cautions should be taken to use wet deposition fluxes of $NH_4^+$ to
extrapolate emissions of $NH_3$.

*Data availability*. Data used in this study are available from the corresponding authors.
*Supplement*. The supplement materials are available online.
*Author contribution*. X. Y. and L. Z. designed the study, analyzed he data and prepared the manuscript.
*Competing interests.* The authors declare that they have no conflict of interest.
*Acknowledgments.* X.Y. is supported by the National Key Research and Development Program in
China (No. 2016YFC0200500), and L.Z. by the Air Pollutants program of Environment and Climate
Change Canada.

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

**List of Figures**

**Figure 1.** Fitting monthly $F_{wet}$ of $SO_4^{2-}$ against the climatology values from every two years using LR with zero interception at Site 1 according to the new approach described in Section 2. Fitted lines represent the LR function with zero interception using 24 elements. x, y and $R^2$ in the legend represent climatology monthly $F_{wet}$, monthly $F_{wet}$ in every two-year and the coefficient of determination in LR analysis, respectively. * reflects the maximum value (cycled markers) excluded for LR analysis and all P values <0.01.

**Figure 2.** m-values and annual $F_{wet}$ of $SO_4^{2-}$, $NO_3^-$ and $NH_4^+$ in 1988-2011 at Site 1, and the annual emissions of $SO_2$ and $NO_x$ in 1990-2011 in Québec and Ontario, Canada. Full and empty markers in blue in (a), (d) and (g) represent the calculation of m-values without and with the outlier, respectively. Empty markers in red represent the outliers in m-values and are excluded for trend analysis, as detailed in Section 2. $R^2$ reflects the coefficient of determination of a variable against the calendar year from LR analysis, and the fitted lines represent the LR function. M-K results are shown in (a-b), (d-e) and (g-h). Phases 1, 2 and 3 in (a) and (c), Phases 1 and 2 in (d) and (f) were gained from PLR presented in Section 3.1.

**Figure 3.** Same as in Fig. 2 except for Site 2, and the annual precipitation and annual emissions in British Columbia, Canada. Horizontal dashes in (b) represent precipitation, and the fitted lines represent the LR function.

**Figure 4.** The mean wind vector and speed (shading area) during 1990-2011 (a), the anomalies of wind vector and wind speed (shading area) during 1990-2001 (b), 2002-2011 (c) and 2007 (d) at 925 hPa over the western coastal Canada and U.S. (the anomalies in b, c and d were conducted relative to the 20-year period of 1990-2009 and the wind vector and wind speed were from the North American Regional Reanalysis (NARR) with a spatial resolution of 32 km by 32 km).

**Figure 5.** Regional m-values at Sites 1, 3 and 4: (a): $SO_4^{2-}$, (b): $NO_3^-$, and (c): $NH_4^+$. $R^2$ reflects the coefficient of determination of a variable against the calendar year from LR analysis, and the fitted lines represent the LR function. M-K

results are shown in (a-c). Phases 1, 2 and 3 are shown in (a) and (c). Phases 1 and 2 in (a) and (b) were gained from PLR presented in Section 3.3.

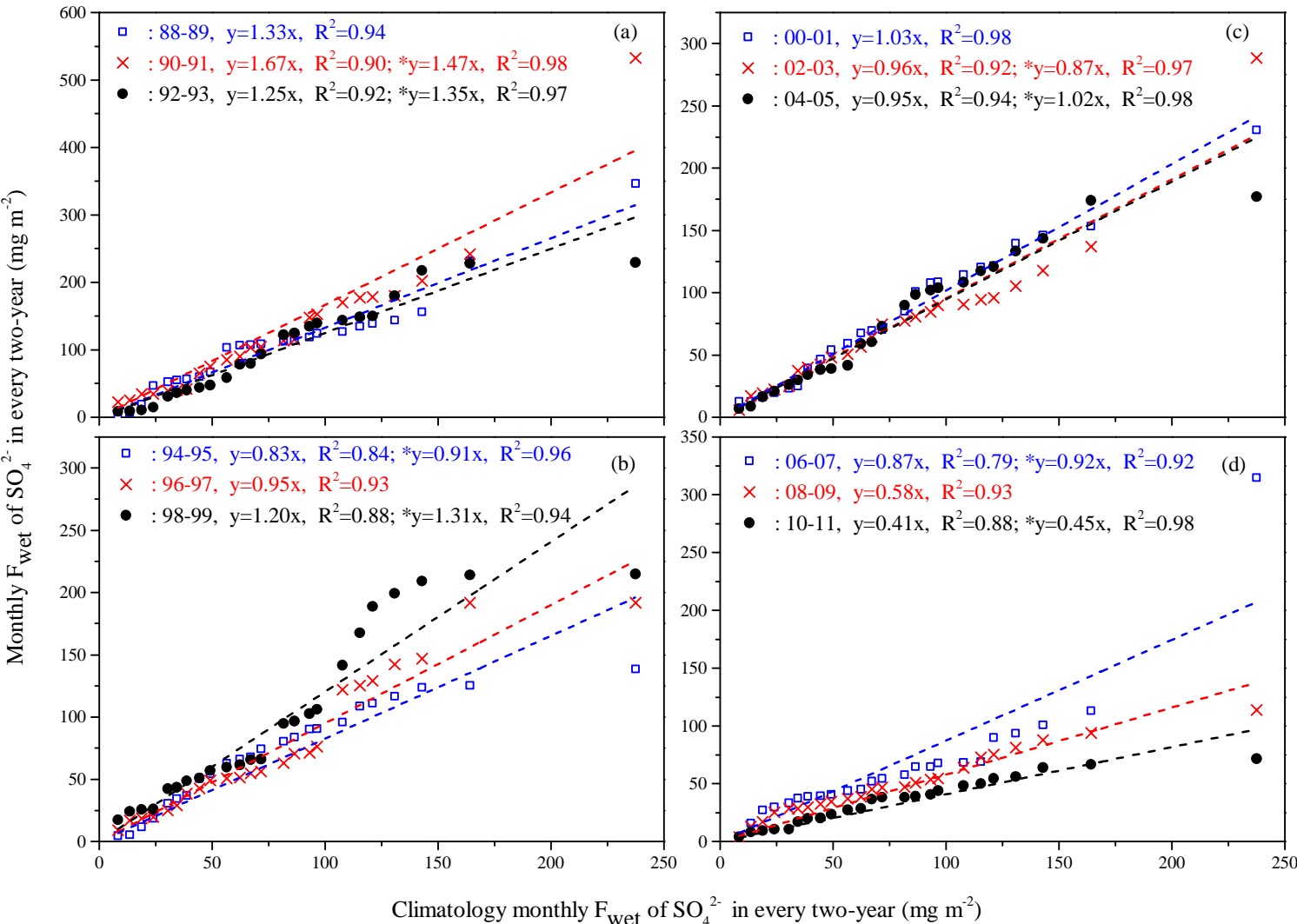

**Figure 1**

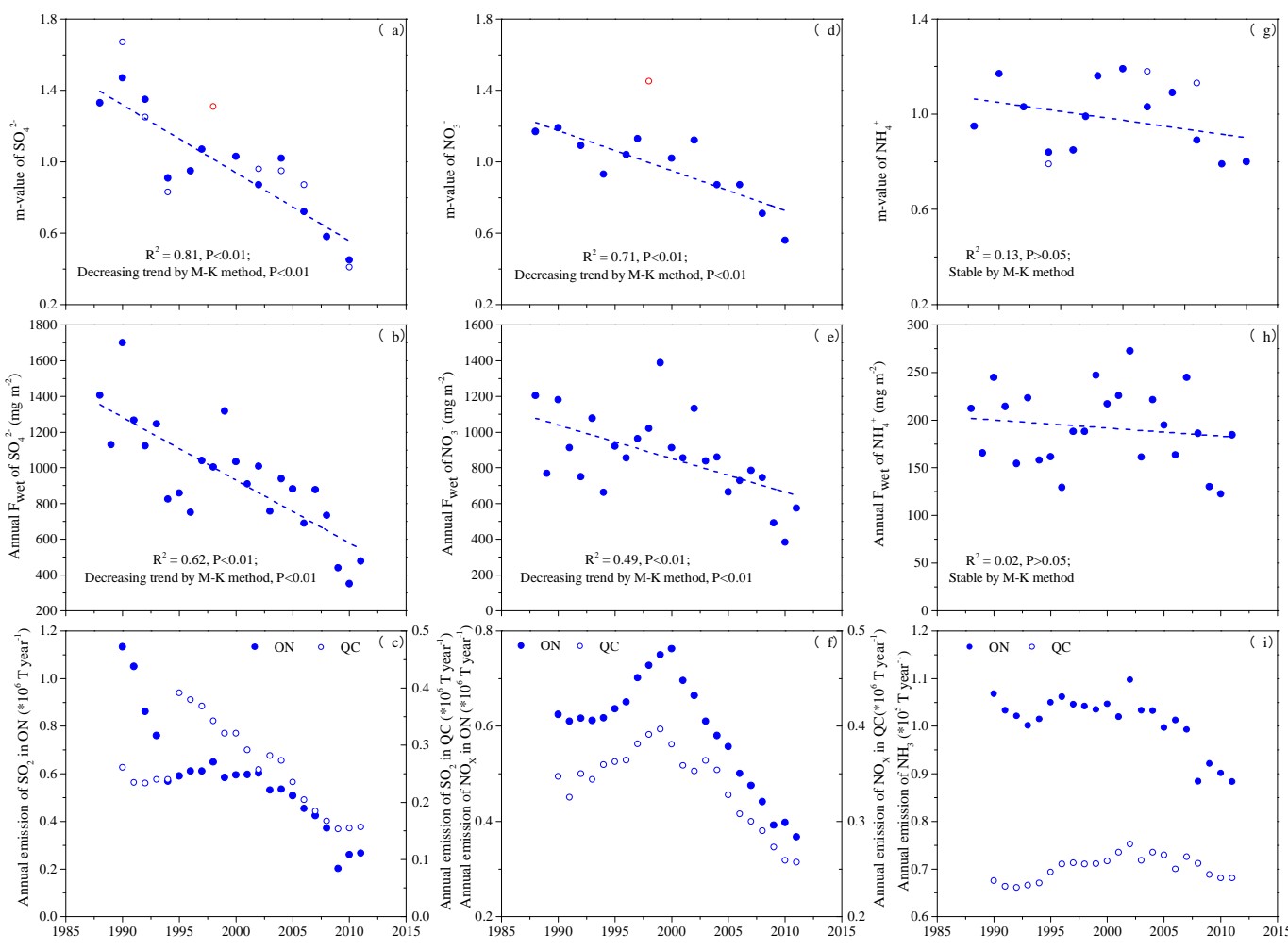

**Figure 2**

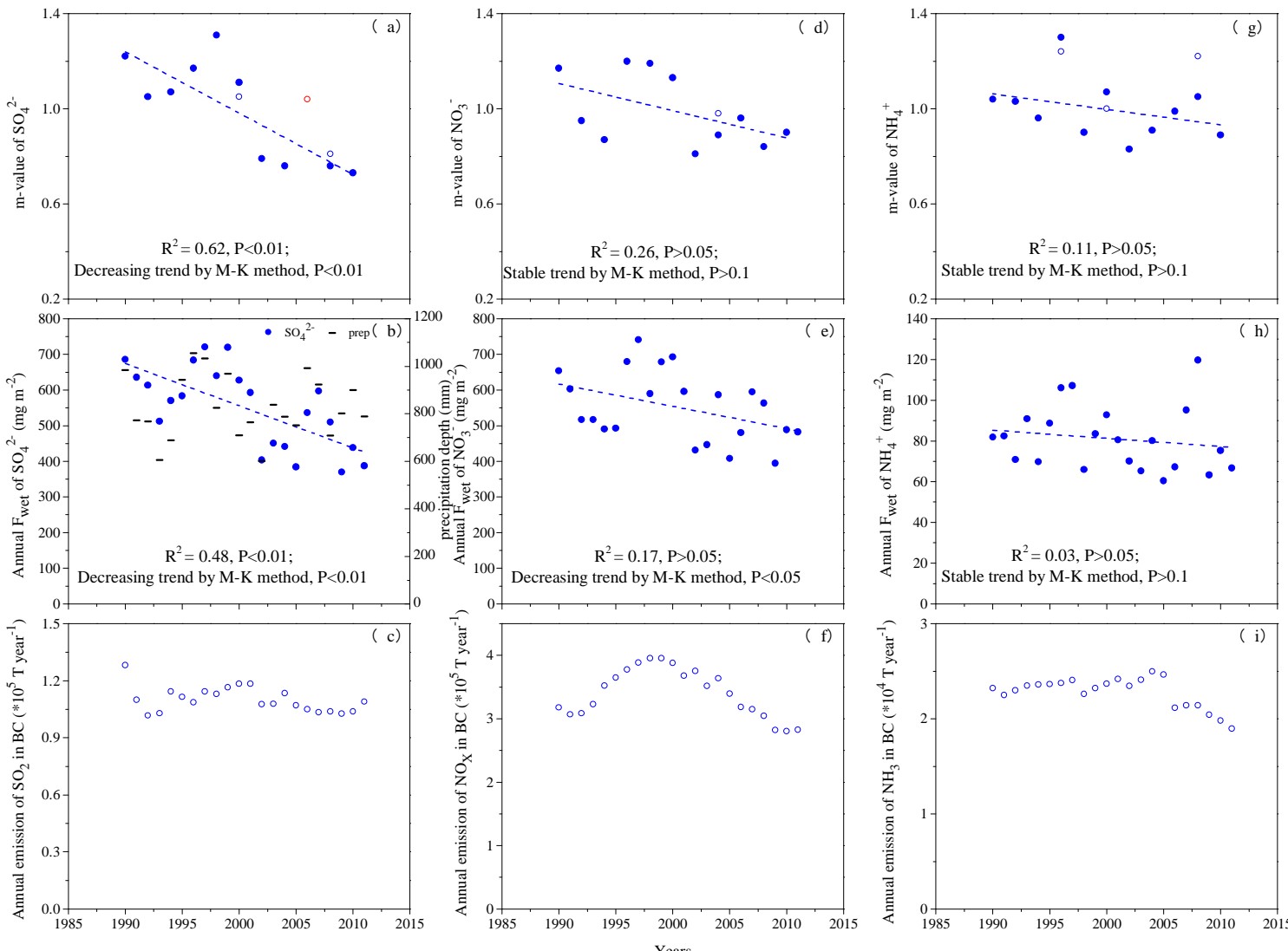

**Figure 3**

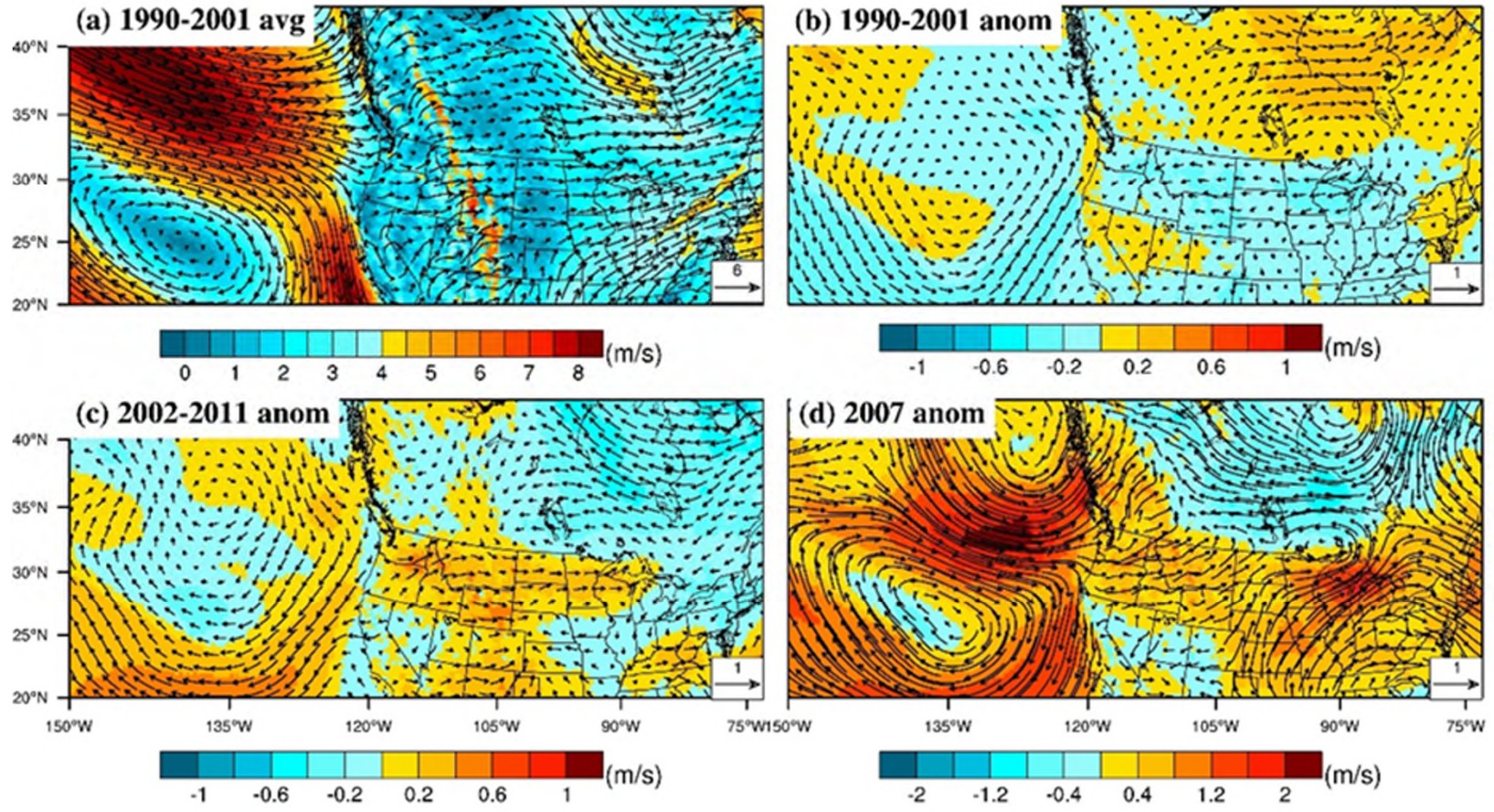

**Figure 4**

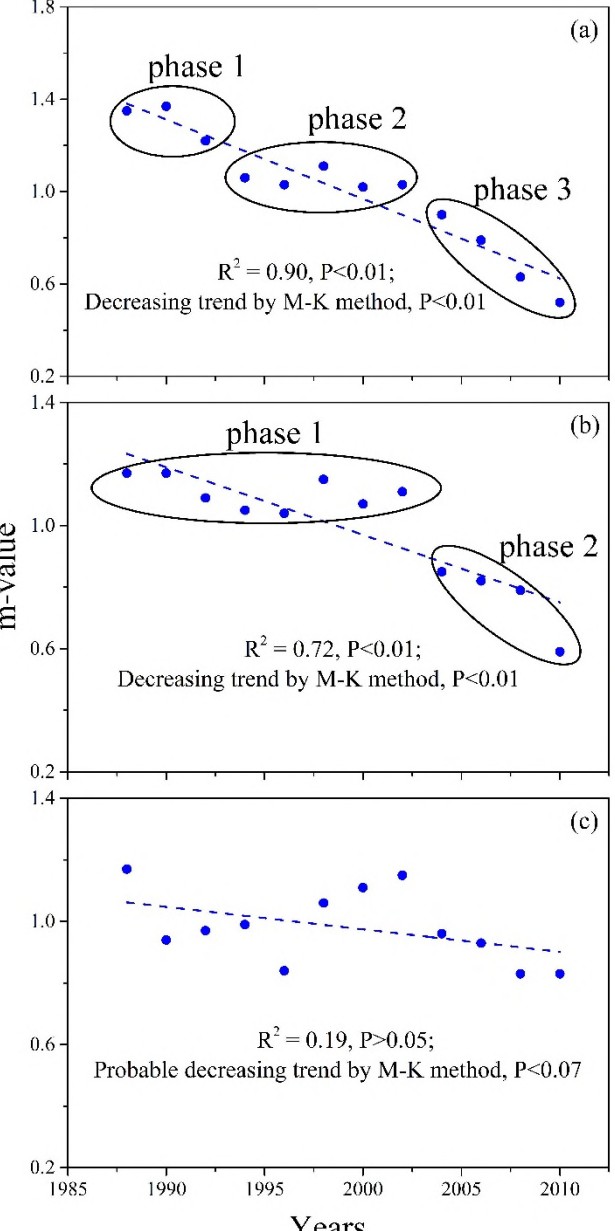

**Figure 5**