# Peer review of "Decoding long-term trends in the wet deposition of sulfate, nitrate and ammonium after reducing the perturbation from climate anomalies"

_Atmospheric Chemistry and Physics, 2019_

## Referee Comment (RC1) · Anonymous Referee #2 · 23 Jul 2019

General comments

This paper asserts that statistical trends analysis of the linkage between emissions changes and measured wet deposition is obscured by multiple factors including climate anomalies. The target analytes of wet deposition measurements ($SO_4^{-2}$, $NO_3^-$, $NH_4^+$) undergo complex atmospheric transformations from their emitted precursors and washout or entrainment in precipitation are dependent on the climate. The climate anomalies are not specifically identified, but evidence exists that they influence relationships between wet deposition and emission trends and are occurring more frequently. Thus, a need exists for a statistical analysis technique to reduce the impact

of the climate anomalies and increase the time interval of comparisons as emission changes, especially those due to regulations, are phased in incrementally and are not linear. The authors propose a statistical method based on the development of an input dataset termed "climatology" (an average of 12 ranked 24-month wet deposition flux measurements) and trends analyses to produce regression slopes for each of the 12 individual 2-year periods considered and the averaged "climatology" dataset. The regressions used are based off the Mann-Kendall (M-K) method, linear regression (LR), and piecewise linear regression (PLR). The authors propose that the time-series of the derived regression slopes better reflects the trends in reported emissions of precursor gases, than the time-series of the annual wet deposition flux data itself.

The method is novel and the m-value time-series relates better to emissions time-series than wet deposition flux (Fwet) time-series at Site 1 for SO4-2 and to a lesser degree for NO3. The m-value time-series appears to reflect inflection points in the emissions time-series that are not as easily observable in the wet deposition flux time-series. However, the method does not improve the relationship of m-values of NH4+ to NH3 emissions at site 1. Furthermore, the method does not seem to show improved m-value correlation with emissions over the annual Fwet data any other location (at Sites 2, 3, and 4) or species. There is no direct comparison metrics to gauge the improvement of the m-values over the annual Fwet other than visual interpretation of plots. The m-value time-series will obviously be visually "cleaner" since a) the m-value has outliers removed and b) the m-value represents 24 datapoints and the annual Fwet represents 12 datapoints.

The largest problem with the study is that that technique is not demonstrated to be robust. The method hinges on the stability of the m-values, but they are very susceptible to the large-value outliers (e.g. example described in text for '90-'91, causes a 0.2 change in m-value; shown in Fig 1). Moreover, for Site 1, the authors acknowledge that 8 of 12 (67%) of datasets needed to have an outlier removed, which from my interpretation greatly compromises the robustness of this technique and its applicability to

different datasets. There appears to be a m-value error analysis conducted with three different approaches in the Supplemental section (Approaches A through C), but no summary or assessment of m-value stability or uncertainty is given. This needs to be developed much more.

Sources of uncertainty in the m-values are not discussed. A reader will likely ask why are large Fwet values so frequently (the 8 of 12 datasets mentioned above) in disagreement with the rest of the monthly values? This question is not answered. What causes the large flux (other than climate anomaly?) Is this a high or low rain event month? Is this rain after a stagnation event?

There is too much assigning uncertainty to vague "Climate anomalies" and "interannual climate variability". These concepts are neither adequately defined nor is any impact that they might have on monthly wet deposition values identified. The section on "interannual climate variability" could be strengthened with local ambient concentrations which are possibly available. At the very least, some more detail and explanation describing the meaning of Fig 4 and how it was derived and its effect on sulfur could be provided.

The reader will also pause as to why so much network-validated data is omitted. Most of the rationale appears to be statistically based (i.e. 'because it doesn't fit the trend'; see the $\pm\,3\sigma$ criteria presented on line 173) which is insufficient without some scientific support (see the discussion on uncertainty of m-values above). More worrisome is the omission of the m-values (i.e. omission of 24 network-validated datapoints) in 1999 on the basis that they don't fit the expected emission trend and are "probably caused by a large perturbation in climate anomalies", but no real evidence is presented.

Specific comments (Individual Science Q)

On page 8, the authors state (line 172) which in turn increase the relative contribution of the air pollutants' emissions to the calculated value. I assume that the authors are presuming that a monthly change in emissions would not impact the Fwet as much as

a large monthly change in precipitation depth or concentration in precipitations. This point should be stressed more in the discussion.

The text does not adequately describe Fig 2 (lines 208 to 213). What is shown and why? I assume the objective of this plot is 1) to show the improvement of the fitted trend of the top row (m-value time-series) to the middle row (Fwet time-series). A metric (correlations with emissions?) is needed to demonstrate the advantage of the m-value over the annual Fwet.

The secondary objective of Fig 2 is to show the incremental trends or "phases". The plots do not currently accomplish this as Phase 1, 2, or 3 are not shown. Also, the PLR segments for Phase 1, 2, and 3 identified in the text are not described. The overall fits shown (e.g. R2 = 0.81 in 2a and R2 = 0.62 in 2b) are not significant in the analysis, but are shown on the plots. The PLR segments should be shown for the emissions as well (or at least compared with the 2a PLR segments).

Considering both of these objectives, the strength in this technique appears to be that the PLR segments for the top-row more closely resemble the PLR segments for the bottom row and that the PLR segments for the middle row do not reflect this. Please reorganize the discussion and analysis to support this. For example, the lines from 283-293 describing the improvement of the m-values over the annual Fwet data should be elaborated on and moved up in the discussion.

I assume the phase year classifications proposed by the authors (Phase 1, 2 and 3) are derived from the emissions data patterns, but the logic behind the years of the phases is not specifically discussed (i.e. why 1988 to 1993 and not 1995?) Do the phases align with emissions regulation implementation?

The PLR segments are often derived from a set of points as low as N=5 (e.g. Phase 1 from 1988 to 1993). Comparisons should state that this is a low N for comparison. On lines 348 -354; the m-value time-series for Site 2 NO3- (Fig 3d) is interpreted to support the decadal shift hypothesis. However, strictly observing the data, without the

hypothesis in mind, it is clear that the four m-values from 1990 and '96-'00 are elevated, while the values from '92 and '94 are similar to values observed after the decadal shift has taken place. This is acknowledged in the text, but no support given other than it is attributable to climate anomalies.

TECHNICAL COMMENTS 1. Figures need descriptive captions and local explanations. 2. Labels on Fig 2 (title incorrect) 3. line 223: "in contrast". Suggest removal, not really in contrast. 4. line 237: Vlaue should read value 5. For Figure 1, distinguish the outlier point removed for each plot (as done in Fig 2) also specify which fit (R2 and p-value applies to the modified fit (I believe it is *, but it is not labeled).

---

## Referee Comment (RC2) · Anonymous Referee #1 · 5 Aug 2019

Review for ACP-2019-418 Decoding long-term trends in the wet deposition of sulfate, nitrate and ammonium after reducing the perturbation from climate anomalies. Xiaohong Yao and Leiming Zhang

General comments

The objective of the study is to understand the effect of emission reduction on long term trends of wet deposition of inorganic ions. In that purpose, the effects of climate anomalies must be isolated to better highlight emission reduction effects. A two decade dataset of wet deposition of $SO_4^{2-}$, $NO_3^-$ and $NH_4^+$ is studied with a new approach at rural Canadian sites. A new method is applied to extract trends and inflection points,

by pre processing the data sets and applying further common statistical tools such as M-K and L-R methods.

The presentation of the new pre processing method based on slopes for monthly wet deposition fluxes during 2 year periods should be clarified and better explained to avoid losing the reader. Indeed, there are several explanations that would need to be better justified to highlight the true added value of this new method.

The summary stipulated that more robust results are found with this new method, but this seems to be only true because some points are excluded from the correlation analysis. The robustness of the method needs further justification. This sentence on robustness in the abstract has to be removed, unless it is really justified.

This analysis is based on the assumption that removing the maximum wet deposition flux corresponds to removing climate anomalies, based on a pre processing of the data: this is exactly the point that has to be better justified, because all the analysis of the results relies on that statement.

When reaching the conclusion, the reader understands that the role of climate anomalies is also very important. The displayed purpose of the paper should be to highlight both the roles of climate anomalies and emission trends, rather than only focusing on emissions. The link with climate anomalies is also an interesting way in understanding the wet deposition flux trends.

The statistical approach lacks from being scientifically justified in terms of geophysical variable influence. I recommend major revision for this study, especially concerning the climate anomalies justification. Indeed, if this part is not well justified, the rest of the study cannot follow.

Generally, a table with a summary of different phases of trends for each site and each ion would help to better capture the results.

Specific comments

Introduction

Wet deposition fluxes of sulfate, nitrate and ammonium are affected by emissions of precursors, atmospheric processes, and climate anomalies. A definition of complex atmospheric processes and climate anomalies that are specifically linked to this study would be useful. Moreover, some more details are expected to explain these three processes, references to literature are not sufficient.

Methodology

This paragraph should be separated into different sub-paragraphs, with 2.1 statistical methods (line 112), 2.2 Data sets (line127), 2.3 Filtering climate anomalies (or something like that, line 152).

Line 125 and below: it is not clear why you use annual wet deposition fluxes as input data, whereas a modified dataset is based on monthly wet deposition fluxes. A figure would be useful to understand how this new dataset is built.

Line 145, what is the scientific explanation of excluding the maximum deposition flux when it deviates from the general regression? You only give a statistical explanation, which does not help in understanding the underlying geophysical causes.

Line 154: do you mean twelve two year periods of data?

Line 159: again, you mention the exclusion of maximum values of wet deposition fluxes, which leads to an increase of the R2 values: this is obvious statistically, but the scientific justification of this exclusion must be clarified. The explanation line 167 that maximum values are believed to be caused by climate anomalies is just a hypothesis and not sufficient to prove that you can exclude this maximum. Furthermore, this paragraph about filtering and excluding values is a bit strange in the methodological section, as it presents results already.

Line 190: Comparisons between this new approach (Approach C) and traditional approaches (A and B) are given in supporting information. Why a 12 month period is

used in approaches A and B, whereas a 24 month period is used in approach C?

Line 195: what do you mean by "a small portion of climate anomalies that are unable to be removed by the new approach"? This is not precise enough.

Results and discussion

Line 208: as mentioned in the general comment, all the analysis of results here relies on the assumption that removing maximum wet deposition fluxes are associated to climate anomalies, which has to be better justified in the methodological section.

Line 210: please specify Fig 2a for $SO_4^{2-}$, 2d for $NO_3^-$ (and so on).

Line 212: where can we check that $NH_4^+$ exhibits a stable trend from M-K analysis, in Fig 2?

From line 215 and below, are you still commenting Fig 2? Please specify to facilitate the reading.

Line 241: the sharp increase in $NO_3^-$ wet deposition flux in 1999 is supposed to be due to a "probable large perturbation from climate anomalies": this is not sufficiently justified. A scientifically argument needs to be provided.

Line 252: "Note that... here" should be declared in the method section, not in the results section. Moreover, R2 are written in the figures, and the text stipulates that R values will be used: this is not consistent.

Line 268: again, perturbations from climate anomalies unable to be removed by the new approach needs to be specified: what can they be exactly? What do they represent in terms of geophysical variables?

Line 282: please detail "many other factors" for describing $NH_4^+$ trends.

Line 293: after comparing m-values and annual deposition fluxes in the paragraph (lines 284-293), what is the interpretation of the statements? What do you want to

highlight here?

Line 301: paragraph 3.2 should be separated into different sub paragraphs (based on ions for example).

Line 388: again, justify which climate anomalies you are talking about to remove m-values

Line 413: what is the reason of unrealistic emission inventory? It could be useful to recall here which emission inventory is used here.

Conclusions

Line 456: this statement about the importance of climate anomalies vs emission trends is really interesting but unfortunately it is not specified earlier as an objective of the study: rather than removing climate anomalies, the purpose of the study could be to highlight the roles of both emission trends and climate anomalies, depending on the periods. The conclusion ends with results consideration that should be in the results section. The conclusion has to be more general and give some general clues for the interpretation of results that were presented. In the present state, it seems that the conclusion is not terminated.

---

## Author Comment (AC1) · 26 Sep 2019

**Response to Referee #1**

We greatly appreciate the reviewer for proving the constructive comments, which have helped us improve the paper quality. We have revised the paper accordingly as detailed in our point-by-point responses below.

RC- Reviewer's Comments; AR – Authors' Responses

*RC: General comments*
*The objective of the study is to understand the effect of emission reduction on long term trends of wet deposition of inorganic ions. In that purpose, the effects of climate anomalies must be isolated to better highlight emission reduction effects. A two decade dataset of wet deposition of $SO_4^{2-}$, $NO_3^-$ and $NH_4^+$ is studied with a new approach at rural Canadian sites. A new method is applied to extract trends and inflection points, by pre processing the data sets and applying further common statistical tools such as M-K and L-R methods. The presentation of the new pre processing method based on slopes for monthly wet deposition fluxes during 2 year periods should be clarified and better explained to avoid losing the reader. Indeed, there are several explanations that would need to be better justified to highlight the true added value of this new method. The summary stipulated that more robust results are found with this new method, but this seems to be only true because some points are excluded from the correlation analysis. The robustness of the method needs further justification. This sentence on robustness in the abstract has to be removed, unless it is really justified. This analysis is based on the assumption that removing the maximum wet deposition flux corresponds to removing climate anomalies, based on a pre processing of the data: this is exactly the point that has to be better justified, because all the analysis of the results relies on that statement. When reaching the conclusion, the reader understands that the role of climate anomalies is also very important. The displayed purpose of the paper should be to highlight both the roles of climate anomalies and emission trends, rather than only focusing on emissions. The link with climate anomalies is also an interesting way in understanding the wet deposition flux trends. The statistical approach lacks from being scientifically justified in terms of geophysical variable influence. I recommend major revision for this study, especially concerning the climate anomalies justification. Indeed, if this part is not well justified, the rest of the study cannot follow. Generally, a table with a summary of different phases of trends for each site and each ion would help to better capture the results.*

AR: In the revised paper, we have added more justification to support our analysis, in particular on the types of climate anomalies (e.g., precipitation depth, wind pattern at local and regional scales) that may cause the abnormality of wet deposition fluxes of ions on monthly and annual scales. We have added the geometric average concentration of ions in precipitation and precipitation depth to reveal the influence of geophysical variables on wet deposition. The two parameters clearly demonstrate that the maximum values of wet deposition fluxes of ions that deviated positively from the general trend

were mainly caused by extreme precipitation events rather than abnormal increase or decrease in geometric average concentration of ions. However, the causes were yet to be identified for the maximum monthly value of Fwet that were negatively deviated from the general trend. This later case has also been stated in the revised paper.

Abnormal increase or decrease in wet deposition of ions associated with climate anomalies at one site does not necessarily mean that the abnormality also occur on a regional scale. This is the case when the data at three sites (Sites 1, 3 and 4 in the same region) were combined together. In such a circumstance, the abnormality identified at a particular site may be a local instead of a regional phenomenon. Thus, the maximum value of wet deposition that deviated substantially from the regression curve needs to be removed for identifying general trends caused by emission trends at one sampling site. Removing the abnormal maximum value of wet deposition would minimize the effects of climate anomalies on the calculated m-values and subsequently derived trend results, thus focusing on the effects of emission control policies. As demonstrated in the revised Supporting Information and revised paper, the new approach proposed in this study is indeed more robust than simply using annual $F_{wet}$ as data input for trend analysis. Following the reviewer's recommendation we have replaced the world "robustly" with "statistically" in several places.

We should not overemphasize the effect of climate anomalies on wet deposition of ions when the data at one site was analyzed just in case it is only a local phenomenon. Moreover, abnormal increase (decrease) in wet deposition of ions due to climate anomalies at one site does not necessarily mean the abnormal increase (decrease) in the total deposition of ions (wet plus dry). Regarding the impacts of atmospheric deposition on eco-systems, the total deposition of ions (wet plus dry) should be more important than wet deposition alone. We prefer to focus on how to removing climate anomalies and to establish the relationship between wet deposition of ions and their corresponding emissions of air pollutants in this study. We agree with the reviewer that it is valuable to compare the effects of climate anomalies on wet deposition at different sites in a regional scale in terms of their similarity and differences, but such effort requires datasets larger than what is available in our study (and is out of the scope of the present study).

Several methods can be used to do PRL analysis in literature. The simplest one is to manually conduct piecewise regression, where inflection points are visibly recognized, and this method is also used in the present study. A few complex algorithms are also available in the literature to conduct PRL if with hundreds of data points (Ryan and Porth, 2007 and references cited there). The complex algorithms are, of course, seldom used to identify trends in annul wet deposition of ions because of the shorter data record history. The reference (Ryan and Porth, 2007) has been added in the revision.

A table summarizing phase classification for m-values of wet deposition of SO42- and NO3- at different sites has been added in the Supporting Information (Table S2).

*RC: Introduction Wet deposition fluxes of sulfate, nitrate and ammonium are affected by emissions of precursors, atmospheric processes, and climate anomalies. A definition of complex atmospheric processes and climate anomalies that are specifically linked to this study would be useful. Moreover, some more details are expected to explain these three processes, references to literature are not sufficient.*

AR: We have revised the second paragraph of Introduction to address this comment, which reads: "The wet deposition of $SO_4^{2-}$, $NO_3^-$ and $NH_4^+$ is affected by not only their gaseous precursors' emissions (Butler et al., 2005; Fowler et al., 2007; Li et al., 2016) but also complex atmospheric processes such as long-range transport, chemical transformation, and dry and wet removal (Cheng & Zhang, 2017; Yao & Zhang, 2012; Zhang et al., 2012). These processes can be largely affected by climate anomalies. For example, climate anomalies can sometimes bring extreme precipitation amounts throughout a particular month, and subsequently lead to extremely high wet deposition fluxes of ions through enhanced wet removal rainout and washout of air pollutants."

*RC: Methodology This paragraph should be separated into different sub-paragraphs, with 2.1 statistical methods (line 112), 2.2 Data sets (line127), 2.3 Filtering climate anomalies (or something like that, line 152).*

AR: The section has been split into subsections: "2.1 Data sources, 2.2 statistical methods, 2.3 Filtering climate anomalies, 2.4 Example case for filtering, 2.5 Justification for the new approach."

*RC: Line 125 and below: it is not clear why you use annual wet deposition fluxes as input data, whereas a modified dataset is based on monthly wet deposition fluxes. A figure would be useful to understand how this new dataset is built.*

AR: Annual wet deposition fluxes are widely used as input data to derive the trend in literature. Annual wet deposition fluxes are the sum of monthly wet deposition fluxes. The newly developed approach in this study discards the simple sum of monthly wet deposition fluxes. Alternatively, we use monthly wet deposition fluxes to derive m-value by removing the abnormal maximum value for trend analysis. In fact, both annual wet deposition fluxes and m-values are based on monthly wet deposition fluxes.

To demonstrate the advantage of our newly developed approach against the conventional approach, we conducted a comparison of their performance in predicting the trend. The clarification has been added in the revised paper, which reads: "The annual Fwet is widely used for trend analysis and the trend results are thereby used to compare with those derived from the approach proposed in this study."

Our example case presents all details while adding new figure may be redundant. Please

see our revised Supporting Information.

RC: Line 145, what is the scientific explanation of excluding the maximum deposition flux when it deviates from the general regression? You only give a statistical explanation, which does not help in understanding the underlying geophysical causes.

AR: Extreme precipitation depth led to the monthly maximum deposition fluxes of ions severely deviating from the general regression. In the revision, it reads as "The actual observed maximum value of 532 mg m$^{-2}$ month$^{-1}$ was much larger than the upper range of the predicted value and was thus believed to be caused by monthly scale climate anomalies, i.e., the occurrence of extreme amount of precipitation. The maximum monthly deposition flux in 1990-1991 occurred in September 1990 when the monthly precipitation depth reached 294 mm, which was much higher than those in the same month of other years, e.g., 169, 68, 95 and 127 mm in 1988, 1989, 1991 and 1992, respectively. The maximum daily precipitation depth in September was also higher in 1990 (91 mm) than in other years (43.6, 12.2, 13.6 and 26.8 mm in 1988, 1989, 1991 and 1992, respectively). However, the monthly geometric average concentration of SO$_4^{2-}$ in precipitation (1.8 mg L$^{-1}$) in September 1990 was close to the mean value (1.7±0.3 mg L$^{-1}$) in September 1988-1992 and was even smaller than that (2.9 mg L$^{-1}$) in August 1990. The maximum value was treated as an outlier and excluded for analysis."

RC: Line 154: do you mean twelve two year periods of data?

AR: corrected.

RC: Line159: again, you mention the exclusion of maximum values of wet deposition fluxes, which leads to an increase of the R2 values: this is obvious statistically, but the scientific justification of this exclusion must be clarified.

AR: Please see our response above to the comment on Line 145.

RC: The explanation line 167 that maximum values are believed to be caused by climate anomalies is just a hypothesis and not sufficient to prove that you can exclude this maximum. Furthermore, this paragraph about filtering and excluding values is a bit strange in the methodological section, as it presents results already.

AR: Please see our response above to the comment on Line 145. Without an example case for filtering data, it is difficult for readers to capture the exact procedure of the new approach. We prefer to keep this part in the Method section. In Results and discussion, we focus on the analysis of trend results.

RC: Line 190: Comparisons between this new approach (Approach C) and traditional approaches (A and B) are given in supporting information. Why a 12 month period is

*used in approaches A and B, whereas a 24 month period is used in approach C?*

AR: The direct comparison between Approach C and the conventional method (using annual $F_{wet}$) is difficult to demonstrate which one is more robust. We thereby compare the results by using 12 month period (Approach A and B), 24 month period (Approach C) and 36 month period (Approach D).

A linear regression analysis result, assuming zero interception and using the m-values calculated from Approach B against the annual $F_{wet}$ data, showed the value of $R^2$ as high as 0.99 (Fig. S4 added in the revision). Thus, the trend result derived from Approach B is exactly the same as that from using the annual wet deposition flux of ion. When we compare the m-values of Approach C with those of Approach B, the conclusion is applicable for the comparison between Approach C and the use of annual $F_{wet}$.

m-values derived from Approach C are more robust than those from Approach B. This is because the use of 24 month data in Approach C instead of 12 month in Approach B allows gaining high $R^2$ values and minimizing uncertainties of m-values. The trend results would be more robust by using m-values from Approach C than by using those from Approach B.

This has been clarified in Supporting Information. More summary of the comparison has also added in Supporting Information.

*RC: Line 195: what do you mean by "a small portion of climate anomalies that are unable to be removed by the new approach"? This is not precise enough.*

AR: This part has been revised to: "climate anomalies that are unable to be removed by the new approach".

*RC: Results and discussion*
*Line 208: as mentioned in the general comment, all the analysis of results here relies on the assumption that removing maximum wet deposition fluxes are associated to climate anomalies, which has to be better justified in the methodological section.*

AR: Please see our response to the comment on Line 145 above. In addition, we have also added more justification, which reads: "The abnormally increased Fwet of $SO_4^{2-}$ in 1999 was mainly because of the increased precipitation depth (1312 mm), which was the largest during 1998-2011 (the annual average precipitation depth excluding 1999 was 1067±86 mm). However, the geometric average concentration of $SO_4^{2-}$ in precipitation in 1999 (1.0 mg L-1) was close to those in the other years, e.g., 0.9 mg L-1 in 1997 and 1998, and 1.0 mg L-1 in 2000. "

*RC: Line 210: please specify Fig 2a for SO42-, 2d for NO3- (and so on).*

AR: The sentences have been revised to: "$SO_4^{2-}$ and $NO_3^-$ showed decreasing trends from a LR analysis, with $R^2$ values of 0.81 and 0.71, respectively, and P values <0.01 (Fig. 2a and 2d). The decreasing trends were also confirmed by the M-K method analysis. $NH_4^+$ exhibited a stable trend from M-K analysis (Fig. 2g), as well as no significant trend with P value >0.05 from LR analysis. The annual $F_{wet}$ of these ions are also shown in Figs. 2b, 2e and 2f and annual emissions of $SO_2$, $NO_x$ and $NH_3$ in Figs. 2c, 2f and 2i, respectively. These data were used to compare and facilitate analysis in terms of identifying inflection points and the advantage of using the m-value over the annual $F_{wet}$, as presented below. "

*RC: Line 212: where can we check that NH4+ exhibits a stable trend from M-K analysis, in Fig 2? From line 215 and below, are you still commenting Fig 2? Please specify to facilitate the reading.*

AR: The sentence has been revised as: "$NH_4^+$ exhibited a stable trend from M-K analysis (Fig. 2g), as well as no significant trend with P value >0.05 from LR analysis."

The sentence in Line 215 has been revised as: "The m-values of $SO_4^{2-}$ and $NO_3^-$ also allowed for statistical identification of trends in different phases supported by annual variations in emissions of $SO_2$ and $NO_x$ (Figs. 2c and 2f) to some extent."

*RC: Line 241: the sharp increase in NO3- wet deposition flux in 1999 is supposed to be due to a "probable large perturbation from climate anomalies": this is not sufficiently justified. A scientifically argument needs to be provided.*

AR: The sentence has been revised to: "The sharp increase in $F_{wet}$ of $NO_3^-$ occurred mainly in 1999, which was probably due to largely increased annual precipitation depth as mentioned in Section 2.4. The analysis was also supported by the geometric average concentration of $NO_3^-$ in precipitation, which was 1.1 mg $L^{-1}$ in 1999, 5% lower than that in 1988 and only 5-10% higher than those in 1990-1991, 1993 and 2002. "

*RC: Line 252: "Note that... here" should be declared in the method section, not in the results section. Moreover, R2 are written in the figures, and the text stipulates that R values will be used: this is not consistent.*

AR: This sentence has been moved to the method section. It now reads: "Note that $R^2$ is conventionally used in LR and PRL. However, r instead of $R^2$ is used in correlation analysis. Thus, $R^2$ and r are used for the two types of analyses in this study, respectively."

It is consistent because LR is conducted for trend analysis. In the text, the correlation analysis of m-values with emissions is presented.

*RC: Line 268: again, perturbations from climate anomalies unable to be removed by the new approach needs to be specified: what can they be exactly? What do they represent in terms of geophysical variables?*

AR: We could not identify the exact cause despite extensive analysis. We thereby have revised the text to: "The contrasting correlation results between the two different periods discussed above implied the complex link between wet deposition of $NO_3^-$ and emissions of $NO_x$. One might assume that the perturbation from climate anomalies might not be fully removed by the new approach for the period of 1990-2003, which overwhelmed the effects of NOx emissions on the trends in m-values of $NO_3^-$. Such a possibility is practically very low since the approach works well for the period of 2002-2011. The contrasting results between these two periods are yet to be explained."

*RC: Line 282: please detail "many other factors" for describing NH4+ trends.*

AR: The sentence has been revised to: "Nearly all of the $NH_4^+$ was associated with $SO_4^{2-}$ and $NO_3^-$ in the atmosphere (Cheng and Zhang, 2017; Teng et al., 2017; Tost et al., 2007; Zhang et al., 2012), and the trends in the m-value of $NH_4^+$ could be affected by many other factors besides $NH_3$ emissions and climate anomalies, e.g., gas-aerosol partitioning and different dry and wet removal efficiencies between $NH_3$ and $NH_4^+$, pH value of wet deposition."

*RC: Line 293: after comparing m-values and annual deposition fluxes in the paragraph (lines 284-293), what is the interpretation of the statements? What do you want to highlight here?*

AR: This part has been split into two parts in the revised paper. The first part reads: "In addition, LR analysis of the annual $F_{wet}$ of $SO_4^{2-}$ revealed a decreasing trend (second row in Fig. 2b). The M-K method analysis also confirmed the decreasing trend with annual $F_{wet}$ as input. However, the three-phase trend in $F_{wet}$ of $SO_4^{2-}$ and related inflection points, identified using the m-values discussed above, were not identified by the t-test when simply using annual $F_{wet}$ data as input. Identifying these inflection points are crucial to assess the effectiveness of environmental policies. The correlation between annual $F_{wet}$ and emission was 0.89 for $SO_4^{2-}$ vs. $SO_2$ in Ontario (P values <0.01), while the corresponding r value was as high as 0.96 between m-value and emission. After reducing the perturbations from climatic factors to the annual $F_{wet}$, a stronger correlation was obtained between $F_{wet}$ and emission. The increased r further solidified the dominant contribution of the long-range transport of air pollutants from Ontario rather than Québec to the wet deposition of $SO_4^{2-}$ at Site 1."

The second part reads: "LR analysis of the annual $F_{wet}$ of $NO_3^-$ revealed a decreasing trend (second row in Fig. 2e), confirmed by the M-K method analysis. However, the two-phase trend in $F_{wet}$ of $NO_3^-$ and related inflection point were not identified by the t-test when simply using annual $F_{wet}$ data as input. The correlations between annual $F_{wet}$

and emission were 0.74-0.76 for $NO_3^-$ vs. $NO_x$ in Québec and Ontario (P values <0.01), while the corresponding r values increased to 0.84-0.85 between m-value and emission. Both the identified inflection point and the stronger correlation between m-value and emission demonstrated the advantage of using the m-value over annual $F_{wet}$ of $NO_3^-$ in trend analysis. "

*RC: Line 301: paragraph 3.2 should be separated into different sub paragraphs (based on ions for example).*

AR: The section has been split into: "3.2.1 Trend in m-value of $SO_4^{2-}$, 3.2.2 Trend in m-value of $NO_3^-$ and $NH_4^+$.

*RC: Line 388: again, justify which climate anomalies you are talking about to remove m values*

AR: This part has been revised to: "The three-phase trend in m-values of $SO_4^{2-}$ and the two-phase trend in m-values of $NO_3^-$ were also obtained at Sites 3 and 4 after excluding a few m-values that were caused by large perturbations from climate anomalies. For example, the annul precipitation depths of 1044 mm in 1987 and 905 mm in 1997 at Site 4 were evidently lower than the average value of 1299±124 mm (excluding 1987 and 1997) in 1985-1997 (Table S2). However, the geometric average concentration of $SO_4^{2-}$ of 1.5 mg $L^{-1}$ in 1997 was the same as the mean value of 1.5±0.2 mg $L^{-1}$ in 1995-1999 (excluding 1997). The value of 1.6 mg $L^{-1}$ in 1987 was also same as that in 1989. The lower annul precipitation depths in 1987 and 1997 than in the other years were very likely the dominant factor causing the abnormally lower m-values in 1986-1987 and 1996-1997."

*RC: Line 413: what is the reason of unrealistic emission inventory? It could be useful to recall here which emission inventory is used here.*

AR: Real on-road emission factors of NOx measured from Transportation and Mobile Equipment in each year of 1990-2003 in Ontario and Quebec are needed to address this issues. Unfortunately, on-road emission factors of NOx are always adopted according to the values in literature rather than measured in different years. Without real on-road emission factors of NOx measured in different years, it is difficult to identify the exact causes. Discussion on emission inventory has been added, which reads: "inconsistent with the bell-shape profile of the NOx emissions mainly caused by annual variations in NOx emission from the sector of Transportation and Mobile Equipment in Ontario and Québec, which could be due to either the perturbation from climate anomalies or unrealistic emissions inventory from (APEI) in Canada."

*RC: Conclusions*
*Line456: this statement about the importance of climate anomaliesvs emission trends is really interesting but unfortunately it is not specified earlier as an objective of the*

*study: rather than removing climate anomalies, the purpose of the study could be to highlight the roles of both emission trends and climate anomalies, depending on the periods. The conclusion ends with results consideration that should be in the results section. The conclusion has to be more general and give some general clues for the interpretation of results that were presented. In the present state, it seems that the conclusion is not terminated.*

AR: Please see our responses to the general comments. We have revised the conclusion accordingly, i.e., removing the detailed results in the second half of the last paragraph in the Conclusion section and make the conclusions more general, which reads: "The long-term variations in Fwet of NH4+ generally showed no clear long-term trends. Moreover, no apparent cause-effect relationships were found between the wet deposition of NH4+ and the emission of NH3. It can be reasonably inferred that additional key factors besides those discussed in this study also impact the trends of Fwet of NH4+. Thus, cautions should be taken to use wet deposition fluxes of NH4+ to extrapolate emissions of NH3."

---

## Author Comment (AC2) · 27 Sep 2019

**Response to Referee #2**

We greatly appreciate the reviewer for proving the constructive comments, which have helped us improve the paper quality. We have revised the paper accordingly as detailed in our point-by-point responses below.

RC- Reviewer's Comments; AR – Authors' Responses

*RC: General comments*
*This paper asserts that statistical trends analysis of the linkage between emissions changes and measured wet deposition is obscured by multiple factors including climate anomalies. The target analytes of wet deposition measurements (SO4-2, NO3-, NH4+) undergo complex atmospheric transformations from their emitted precursors and washout or entrainment in precipitation are dependent on the climate. The climate anomalies are not specifically identified, but evidence exists that they influence relationships between wet deposition and emission trends and are occurring more frequently. Thus, a need exists for a statistical analysis technique to reduce the impact of the climate anomalies and increase the time interval of comparisons as emission changes, especially those due to regulations, are phased in incrementally and are not linear. The authors propose a statistical method based on the development of an input dataset termed "climatology" (an average of 12 ranked 24-month wet deposition flux measurements) and trends analyses to produce regression slopes for each of the 12 individual 2-year periods considered and the averaged "climatology" dataset. The regressions used are based off the Mann-Kendall (M-K) method, linear regression (LR), and piecewise linear regression (PLR). The authors propose that the time-series of the derived regression slopes better reflects the trends in reported emissions of precursor gases, than the time-series of the annual wet deposition flux data itself.*

*The method is novel and them-value time-series relates better to emissions time-series than wet deposition flux (Fwet) time-series at Site 1 for SO4-2 and to a lesser degree for NO3. The m-value time-series appears to reflect inflection points in the emissions time-series that are not as easily observable in the wet deposition flux time-series. However, the method does not improve the relationship of m-values of NH4+ to NH3 emissions at site1. Furthermore, the method does not seem to show improved m-value correlation with emissions over the annual Fwet data any other location (at Sites 2, 3, and 4) or species. There is no direct comparison metrics to gauge the improvement of the m-values over the annual Fwet other than visual interpretation of plots. The m value time-series will obviously be visually "cleaner" since a) the m-value has outliers removed and b) the m-value represents 24 data points and the annual Fwet represents 12 data points.*

AR: We have added the comparison at Sites 3 and 4 in the revised Supporting Information, which reads: "Using the m-values over the annual $F_{wet}$ of $SO_4^{2-}$ improves the r value from 0.73 to 0.87 at Site 3 and from 0.91 to 0.93 at Site 4. Using the mvalues over annual $F_{wet}$ of $NO_3^-$ improves the r value from 0.81 to 0.87 at Site 3 and from 0.78 to 0.89 at Site 4." No significant correlation of m-value and $F_{wet}$ with the corresponding emissions existed at Site 2 and the comparison is thereby not presented.

*RC: The largest problem with the study is that that technique is not demonstrated to be robust. The method hinges on the stability of the m-values, but they are very susceptible to the large-value outliers (e.g. example described in text for '90-'91, causes a 0.2 change in m-value; shown in Fig 1). Moreover, for Site 1, the authors acknowledge that 8 of 12 (67%) of datasets needed to have an outlier removed, which from my interpretation greatly compromises the robustness of this technique and its applicability to different datasets.*

AR: We originally only explained the method from a statistical analysis consideration, which may hinder the real advantage of the method. In the revised paper, we have made substantial revisions in several sections to clarify this point. For example, we have changed this sentence "The actual observed maximum value of 532 mg m$^{-2}$ month$^{-1}$ was much larger than the upper range of the predicted value and was thus believed to be caused by monthly scale climate anomalies" to this: "The actual observed maximum value of 532 mg m$^{-2}$ month$^{-1}$ was much larger than the upper range of the predicted value and was thus believed to be caused by monthly scale climate anomalies, i.e., the occurrence of extreme amount of precipitation. The maximum monthly deposition flux in 1990-1991 occurred in September 1990 when the monthly precipitation depth reached 294 mm, which was much higher than those in the same month of other years, e.g., 169, 68, 95 and 127 mm in 1988, 1989, 1991 and 1992, respectively. The maximum daily precipitation depth in September was also higher in 1990 (91 mm) than in other years (43.6, 12.2, 13.6 and 26.8 mm in 1988, 1989, 1991 and 1992, respectively). However, the monthly geometric average concentration of $SO_4^{2-}$ in precipitation (1.8 mg L$^{-1}$) in September 1990 was close to the mean value (1.7±0.3 mg L$^{-1}$) in September 1988-1992 and was even smaller than that (2.9 mg L$^{-1}$) in August 1990." There are several other similar changes which can be found from the track change version of the paper.

In our approach, only the maximum value in 24 months severely deviated from the general trend was removed to calculate m-values. Thus, we have 95%-100%, i.e., 23/24 -24/24 monthly values, data to calculate m-value with high $R^2$ values (e.g., 0.92-0.98 at Site 1 for $SO_4^{2-}$). The calculated m-value would fully reflect the contribution from emissions of air pollutants since only 5% data are sometimes removed. When the data size is even larger, e.g., the group of Sites 1, 3 and 4, 100% data are used to calculate m-value. Using m-values calculated from Approach C is applicable for different datasets.

Compared with the calculated m-values from Approach B using 12 month data, the use of 24 month data in each array in Approach C largely increased $R^2$ value and decreased uncertainties of the calculated m-values. However, a linear regression analysis result,

assuming zero interception and using the m-values calculated from Approach B against the annual $F_{wet}$ data, showed the value of $R^2$ as high as 0.99 (Fig. S4 added in the revision). This means that the trend analysis results would be the same regardless of using annual $F_{wet}$ data or the m-values as input if Approach B is used. The extracted trends would include larger perturbations from climate anomalies in Approach B. Thus, it can be concluded that the trend analysis results derived from m-values calculated from Approach C would be more robust than those derived from m-values calculated from Approach B. It is also safe to say that the trend results derived from m-values calculated from Approach C are more robust than those derived from annual $F_{wet}$ data.

From Comment 4 listed below, we realize that the original text may mislead the reviewer, i.e., 8/12 datasets needed to have an outlier removed because of the maximum $F_{wet}$ being positively deviated from the general trend. This is of course impossible and may make the reviewer doubt the robustness of Approach C. We have therefore clarified this in the revised paper, which reads: "Three out of the 12 data sets showed the maximum $F_{wet}$ being positively deviated from the general trend, five negatively deviated from the general trend, and four consistent with the general trend."

*RC: There appears to be a m-value error analysis conducted with three different approaches in the Supplemental section (Approaches A through C), but no summary or assessment of m-value stability or uncertainty is given. This needs to be developed much more. Sources of uncertainty in the m-values are not discussed.*

AR: We have added such analysis, and details can be found in section 1 of the revised Supporting Information.

*RC: A reader will likely ask why are large Fwet values so frequently (the 8 of 12 datasets mentioned above) in disagreement with the rest of the monthly values? This question is not answered. What causes the large flux (other than climate anomaly?) Is this a high or low rain event month? Is this rain after a stagnation event?*

AR: The large $F_{wet}$ value was mainly caused by extreme precipitation depth in monthly scale. We have clarified the frequency of the large values (see the response to the comment above). The distribution result is quite normal. Although the maximum monthly value of $F_{wet}$ positively deviated from the general trend was clearly attributed to extreme precipitation, the cause was yet to be identified for the maximum monthly value of $F_{wet}$ negatively deviated from the general trend. This latter case has also been stated in the revised paper.

*RC: There is too much assigning uncertainty to vague "Climateanomalies" and "interannual climate variability". These concepts are neither adequately defined nor is any impact that they might have on monthly wet deposition values identified. The section on "interannual climate variability" could be strengthened with local ambient concentrations which are possibly available.*

AR: We have revised discussions where appropriate throughout the paper. For example, the secondary paragraph of Section 3.2.2 has been revised substantially, which now reads: "In addition to decadal anomalies of wind fields, the interannual climate variability such as precipitation depth, annual anomalies of wind fields in 2007, etc., (Fig. 3b) also affected the trends in m-values and annual $F_{wet}$ of $NO_3^-$. The annual precipitation depth largely varied from 601 mm to 1054 mm in the two decades. The perturbations from interannual variability of precipitation depth cannot be completely removed by the new approach. For example, the calculated m-values in 1992-1993 and 1994-1995 were evidently lower than the m-values in 1990-2001. However, the annual geometric average concentrations of $NO_3^-$ in 1992-1995 varied around 0.77±0.11 mg $L^{-1}$ and were even larger than the values of 0.66±0.08 mg $L^{-1}$ in 1990-2001 (excluding 1992-1995). The lower m-values were mainly attributed to the lower precipitation depth in 1992-1994 (Fig 3b) rather than lower emissions of NOx. Interannual climate variability including precipitation depth and annul anomalies of wind fields may complicate the relationship between the $F_{wet}$ of $NO_3^-$ and the emissions of $NO_x$ in British Columbia." Also in the Conclusion section, the revised version on this point reads: "At this location, the decreasing trends in $F_{wet}$ of $SO_4^{2-}$ and $NO_3^-$ were caused by the decadal anomalies of wind fields, as well as being affected by interannual climate variability including precipitation depth and annul anomalies of wind fields, etc., which overwhelmed the impact of the emission changes of the gaseous precursors in this province. This is the first study that has identified that decadal anomalies of wind fields can dominate trends in $F_{wet}$ of $SO_4^{2-}$ and $NO_3^-$."

RC: At the very least, some more detail and explanation describing the meaning of Fig 4 and how it was derived and its effect on sulfur could be provided.

AR: In Fig. 4, the re-analysis data are used. The re-analysis data have been constrained by observational data and the reference has been cited. We have also added more detailed discussion, which reads: "The wind vector and wind speed from the North American Regional Reanalysis (NARR), with a spatial resolution of 32 km by 32 km (Mesinger et al., 2006), were thereby analyzed to study the decadal changes in wind fields and associated potential impacts on the long-range transport of air pollutants over the western coastal Canada and U.S. The average wind fields including mean wind vector and speed (shading in Fig 4a-d) in 1990-2011 at 925 hPa showed air masses over the western coastal Canada and U.S. were primarily originated from the Pacific Ocean (Fig. 4a). However, the anomalies of wind fields in 1990-2001 relative to 1990-2009 clearly showed a counterclockwise pattern in the corresponding coastal area, including Site 2., while a clockwise pattern existed in 2002-2011 relative to 1990-2009 (Fig. 4b, c). The anomalies shown in Fig. 4c indicated the northwesterly wind being enhanced in 2002-2011 over the western coastal Canada and U.S., possibly reducing air pollutants being transported from the continent to Site 2. In contrast, the anomalies in Fig. 4b indicated that the northwesterly wind was reduced in 1990-2001. Consequently, more air pollutants might have been transported from the continent to Site 2, resulting in a

distinct demarcation in 2002. This hypothesis was also supported by a large rebound of the m-value in 2006-2007, due to the increase in $F_{wet}$ of $SO_4^{2-}$ in 2007. The climate anomalies of wind fields in 2007 relative to 1990-2009 showed a counterclockwise pattern in the north, while the clockwise pattern was pushed to the south (Fig. 4d). With the northwesterly wind being reduced, a greater contribution of air pollutants from the coast of Canada and U.S. to Site 2 might have led to the large increase in $F_{wet}$ of $SO_4^{2-}$ during a few month-long periods in 2007."

*RC: The reader will also pause as to why so much network-validated data is omitted. Most of the rationale appears to be statistically based (i.e. 'because it doesn't fit the trend'; see the ±3σ criteria presented online 173) which is insufficient without some scientific support (see the discussion on uncertainty of m-values above). More worrisome is the omission of the m-values (i.e. omission of 24 network-validated datapoints) in 1999 on the basis that they don't fit the expected emission trend and are "probably caused by a large perturbation in climate anomalies", but no real evidence is presented.*

AR: The evidence has been added in the revision in a few places, (1) "The abnormally increased $F_{wet}$ of $SO_4^{2-}$ in 1999 was mainly because of the increased precipitation depth (1312 mm), which was the largest during 1998-2011 (the annual average precipitation depth excluding 1999 was 1067±86 mm). However, the geometric average concentration of $SO_4^{2-}$ in precipitation in 1999 (1.0 mg $L^{-1}$) was close to those in the other years, e.g., 0.9 mg $L^{-1}$ in 1997 and 1998 and 1.0 mg $L^{-1}$ in 2000." (2) "The sharp increase in $F_{wet}$ of $NO_3^-$ occurred mainly in 1999, which was probably due to largely increased annual precipitation depth as mentioned in Section 2.4. The analysis was also supported by the geometric average concentration of $NO_3^-$ in precipitation, which was 1.1 mg $L^{-1}$ in 1999, 5% lower than that in 1988 and only 5-10% higher than those in 1990-1991, 1993 and 2002."

*RC: Specific comments (Individual Science Q) On page 8, the authors state (line 172) which in turn increase the relative contribution of the air pollutants' emissions to the calculated value. I assume that the authors are presuming that a monthly change in emissions would not impact the Fwet as much as a large monthly change in precipitation depth or concentration in precipitations. This point should be stressed more in the discussion.*

AR: Yes, monthly change in emissions should not impact the $F_{wet}$ as much as large monthly changes in precipitation depth or concentration in precipitations. In the revision, we have added this statement: "Note that monthly changes in emissions may not impact the $F_{wet}$ as much as does a large monthly change in precipitation depth or concentration in precipitation. For example, the monthly average concentrations of $SO_2$ were almost the same in May, September and October of 1990 (~0.7 µg $m^{-3}$) while the monthly $F_{wet}$ of $SO_4^{2-}$ varied significantly, e.g., 113, 179 and 532 mg $m^{-2}$ $month^{-1}$, respectively in the same months. The monthly average concentration of $SO_2$ in February

(4.8 µg m$^{-3}$) was the largest among the twelve months of 1990, but the corresponding monthly $F_{wet}$ of $SO_4^{2-}$ was the smallest (34 mg m$^{-2}$ month$^{-1}$).”

As show in the revised Fig S2c, the geometric average concentrations of $SO_4^{2-}$ at Site 1 in six months of 1996, including February, April, May, June, September and November, narrowly varied around 0.63±0.05 mg L$^{-1}$ (Fig. S2c). The six months were almost evenly distributed in 12 months of 1996. This also suggests that monthly change in emissions would not impact the monthly geometric average concentrations of $SO_4^{2-}$. However, the geometric averages largely oscillated from 0.27 mg L$^{-1}$ to 1.77 mg L$^{-1}$ in the other six months of 1996 at the site. Based on the narrow variation in the former six months, it can be inferred that the large oscillation in the latter six months were less likely due to monthly changes in emissions. For example, the value of 1.77 mg L$^{-1}$ in March of 1996 was the largest and approximately two and half times of 0.68 mg L$^{-1}$ in February of 1996. The monthly average concentrations of $SO_2$ in ambient air were close to each other, i.e., 2.6 µg m$^{-3}$ in March and 2.4 µg m$^{-3}$ in February of 1996. Thus, the large oscillation in the latter six months were very likely due to the effects of climate anomalies imposing on atmospheric processes. However, we cannot quantify what types of climate anomalies caused this. We have added clarification in the revised manuscript and Supporting Information on this point.

*RC: The text does not adequately describe Fig 2 (lines 208 to 213). What is shown and why? I assume the objective of this plot is 1) to show the improvement of the fitted trend of the top row (m-value time-series) to the middle row (Fwet time-series). A metric (correlations with emissions?) is needed to demonstrate the advantage of the m-value over the annual Fwet. The secondary objective of Fig 2 is to show the incremental trends or "phases". The plots do not currently accomplish this as Phase 1, 2, or 3 are not shown. Also, the PLR segments for Phase 1, 2, and 3 identified in the text are not described. The overall fits shown (e.g. R2 = 0.81 in 2a and R2 = 0.62 in 2b) are not significant in the analysis, but are shown on the plots. The PLR segments should be shown for the emissions as well (or at least compared with the 2a PLR segments). Considering both of these objectives, the strength in this technique appears to be that the PLR segments for the top-row more closely resemble the PLR segments for the bottom row and that the PLR segments for the middle row do not reflect this. Please reorganize the discussion and analysis to support this. For example, the lines from 283-293 describing the improvement of the m-values over the annual Fwet data should be elaborated on and moved up in the discussion.*

AR: We have made a substantial revision by reorganizing the discussion and analysis in Section 3.1. Three phases have been labeled in Fig 2. The objectives of Fig. 2 have also been added in the context. We agree that the added objectives makes the part more readable.

The overall fits (e.g. $R^2$ = 0.81 in 2a and $R^2$ = 0.62 in 2b with P<0.01) shown here are significant, i.e., "$SO_4^{2-}$ and $NO_3^-$ showed decreasing trends from a LR analysis, with $R^2$

values of 0.81 and 0.71, respectively, and P values <0.01 (Fig. 2a and 2d)."

*RC: I assume the phase year classification sproposed by the authors(Phase 1,2 nd 3) are derived from the emissions data patterns, but the logic behind the years of the phases is not specifically discussed (i.e. why 1988 to 1993 and not 1995?) Do the phases align with emissions regulation implementation? The PLR segments are often derived from a set of points as low as N=5 (e.g. Phase 1 from 1988 to 1993). Comparisons should state that this is a low N for comparison.*

AR: The three phases of $SO_4^{2-}$ and two phases of $NO_3^-$ were firstly visibly identified by simple screening. We then confirmed the phase results by t-test statistically. This is the simplest way to do PRL analysis if the data size is not too large. The phases were supported by emissions of $SO_2$ and NOx to some extent, but a few inconsistences still existed, e.g., the almost constant m-value of $NO_3^-$ in Phase 1 against the bell-shape distribution of NOx emission in the same Phase.

We don't think that emission data alone can allow classifying these phases of $SO_4^{2-}$ and $NO_3^-$. It is well known that real emissions of air pollutants may not always align with emission regulation schedules. Emissions regulation implementation always needs to be examined by using long-term field measurements. However, emission data can facilitate the analysis of phase changes in m-values, since inflection points of different phases of m-values and emissions should be close to each other.

The m-values in 1988-1993 oscillated approximately 1.38±0.08 while the m-value in 1994-1995 largely decreased down to 0.91, the latter period was clearly related to Phase 2 (1994 to 2005) with m-values around 1.02±0.08. The statistical results confirmed the classification.

The sentence has been revised as: "The m-values of $SO_4^{2-}$ and $NO_3^-$ also allowed for the visible and statistical identification of trends in different phases in support by annual variations in emissions of $SO_2$ and $NO_x$ (Fig. 2c and 2f) to some extent."

In the revision, we have also added: "The three phases generally aligned with the three-phase regulated $SO_2$ emissions in Ontario. It should be stated that Phase 1 and Phase 3 each covered only six years (N=6), respectively. Cautions should be taken to explain the trend result in each phase in relation to precursors' emissions."

*RC: On lines 348 -354; the m-value time-series for Site 2 NO3- (Fig 3d) is interpreted to support the decadal shift hypothesis. However, strictly observing the data, without the hypothesis in mind, it is clear that the fourm-values from 1990 and '96-'00 are elevated, while the values from '92 and '94 are similar to values observed after the decadal shift has taken place. This is acknowledged in the text, but no support given other than it is attributable to climate anomalies.*

AR: In the revision, we have added this statement: "For example, the calculated m-values in 1992-1993 and 1994-1995 were evidently lower than the m-values in 1990-2001. However, the annual geometric average concentrations of $NO_3^-$ in 1992-1995 varied around $0.77\pm0.11$ mg $L^{-1}$ and were even larger than the values of $0.66\pm0.08$ mg $L^{-1}$ in 1990-2001 (excluding 1992-1995). The lower m-values were mainly attributed to the lower precipitation depth in 1992-1994 (Fig 3b) rather than lower emissions of NOx."

RC: TECHNICAL COMMENTS 1. Figures need descriptive captions and local explanations. 2. Labels on Fig 2 (title incorrect) 3. line 223: "in contrast". Suggest removal, not really in contrast. 4. line 237: Vlaue should read value 5. For Figure 1, distinguish the outlier point removed for each plot (as done in Fig 2) also specify which fit (R2 and p-value applies to the modified fit (I believe it is *, but it is not labeled).

AR: Figure captions and labels have all been revised as suggested.

---

## Referee Report (RR1)

Overall, I found the author revisions to clarify the motivation, the method, and the results substantially.
The specific objective was to create a new variable of regression slopes (two-year averages vs. the base average over the whole period (or climatology)) to reduce scatter in monthly wet deposition data to help elucidate patterns in trends.

These reduced slope patterns were used to identify inflection points to break the overall trend into 3 phases (for sulfate at 3 sites in central and eastern Canada) or 2 phases (for nitrate in central and eastern Canada) and link them to patterns in central and eastern Canadian emissions of gaseous precursors ($SO_2$ and NOx). No linkage to emissions could be found for deposited $NH_4$ and gaseous $NH_3$, which was attributed to the complex phase-transitioning, and transport and removal mechanisms.
Additionally, no linkage could be found between wet deposition and emissions in the western Canadian site. An explanation as a decadal wind field shift was offered as a possible explanation for this.

The sections on Climate anomalies have been clarified and defined sufficiently.

Concerning the 'Justification of Data removal and the Applicability of method', there were significant concerns with the removal of data by both reviewers. This included a) the removal of maximum monthly wet deposition flux data and b) the omission of the m-values for 1998-1999. The primary concern is that this compromised the "robustness" of the technique and the applicability to other scenarios.

a) The edits made by the author have made the process and decisions behind the removal of the monthly maximum wet deposition flux data clear. The statistical justification is already clearly explained, and the scientific justification has been expanded to include both the geometric mean of the concentration in the precipitation samples and the precipitation depth. The authors need to further revise this justification to include the washout dilution effect concentrations in precipitation tend to decrease with elevated amounts of sustained precipitation (see Schichtel et al., 2019 STOTEN https://doi.org/10.1016/j.scitotenv.2019.06.104 and references therein; see explanation in 2.2.2 with Washout equation #4 and Figure 2). This example includes several sources with a conflicting interpretation of the justification given by the authors and may also explain the observed negative bias of the 5 of 12 monthly maximum wet deposition flux data which the authors did not address (lines 218-221).

b) I still question if the removal of the 1998-1999 m-value was driven by the desired fit with the sulfate emissions in ON. The nitrate m-values for the same timeframe is elevated along with NOx emissions and its slope (m=1.31) is comparable with those as recent as 1992-1993 (m=1.35). However, the regression of 1998-1999 does appear to be skewed by (N=6) $F_{wet}$ values falling between 100 and 175 mg m$^{-2}$, and there is scientific justification of the geometric mean of the concentration in the precipitation samples and the precipitation depth given for this and in Figure S6. The statistical justification also appears reasonable. While I may disagree with this decision, it is clearly defended and I don't think that it detracts from the applicability of the method and the authors did remove the term "robustly" from descriptions of the technique. No further action is needed on this.

In determining concentration and emission linkages, no uncertainties with the emissions data was identified, except briefly on line 564 (for NOx). The uncertainty of $NH_3$ emissions inventories (dispersion of localized sources, poor characterization of process-level emission) also likely play a role in the lack of correlation with deposited $NH_4$.

With further revision, the section in the Supplemental Information on the comparison of the four approaches (A, B, C, D) is informative and helps to see why the authors chose the method they are proposing.

*TECHNICAL COMMENTS*
- *Figures S5 and S6 should be updated to S7 and S8 (line 523).*
- *Line 526 should read 'annual'*